# Diurnal variability of atmospheric $O_2$, $CO_2$ and their exchange ratio above a boreal forest in southern Finland

Kim A.P. Faassen[1], Linh N.T. Nguyen[2], Eadin R. Broekema[2], Bert A.M. Kers[2], Ivan Mammarella[3], Timo Vesala[3,4], Penelope A. Pickers[5], Andrew C. Manning[5], Jordi Vilà-Guerau de Arellano[1,6], Harro A.J. Meijer[2], Wouter Peters[1,2], and Ingrid T. Luijkx[1]

[1]Meteorology and Air Quality, Wageningen University and Research, Wageningen, the Netherlands
[2]University of Groningen, Centre for Isotope Research, Energy and Sustainability Research Institute Groningen, Groningen, the Netherlands
[3]Institute for Atmospheric and Earth System Research (INAR) / Physics, Faculty of Science, University of Helsinki, Helsinki, Finland
[4]INAR/Forest Sciences, Faculty of Agriculture and Forestry, University of Helsinki, Helsinki, Finland
[5]Centre for Ocean and Atmospheric Sciences, School of Environmental Sciences, University of East Anglia, Norwich, NR4 7TJ, United Kingdom
[6]Atmospheric Chemistry Department, Max Planck Institute for Chemistry, 55128 Mainz, Germany

**Correspondence:** Kim A.P. Faassen (kim.faassen@wur.nl)

**Abstract.** The exchange ratio (ER) between atmospheric $O_2$ and $CO_2$ is a useful tracer of better understanding the carbon budget on global and local scale. The variability of ER (in mol $O_2$ per mol $CO_2$) between terrestrial ecosystems is not well-known, and there is no consensus on how to derive the ER signal of an ecosystem, as there are different approaches available, either based on concentration ($ER_{atmos}$) or flux measurements ($ER_{forest}$). In this study we measured atmospheric $O_2$ and $CO_2$ concentrations at two heights (23 m and 125 m) above the boreal forest in Hyytiälä, Finland. Such measurements of $O_2$ are unique and enable us to potentially identify which forest carbon loss and production mechanisms dominate over various hours of the day. We found that the $ER_{atmos}$ signal at 23 m represents next to the diurnal cycle of the forest exchange also represents other factors, including entrainment of air masses before midday, with different thermodynamic and atmospheric composition characteristics in the atmospheric boundary layer. To derive $ER_{forest}$ we infer $O_2$ fluxes using multiple theoretical and observation-based micro-meteorological formulations to determine the most suitable approach. Our resulting $ER_{forest}$ shows a distinct difference in behaviour between daytime ($0.92 \pm 0.17$ mol/mol) and nighttime ($1.03 \pm 0.05$ mol/mol). These insights demonstrate the diurnal variability of different ER signals above a boreal forest and we also confirmed that the signals of $ER_{atmos}$ and $ER_{forest}$ can not be used interchangeably. Therefore, we recommend measurements on multiple vertical levels to derive $O_2$ and $CO_2$ fluxes for the $ER_{forest}$ signal, instead of a single level time series of the concentrations for the $ER_{atmos}$ signal. We show that $ER_{forest}$ can be further split into specific signals for respiration ($1.03 \pm 0.05$ mol/mol) and photosynthesis ($0.96 \pm 0.12$ mol/mol). This estimation allows us to separate the Net Ecosystem Exchange (NEE) into Gross Primary Production (GPP) and Total Ecosystem Respiration (TER), giving comparable results to the more commonly used eddy covariance approach. Our study shows the potential of using atmospheric $O_2$ as an alternative and complementary method to gain new insights into the different $CO_2$ signals that contribute to the forest carbon budget.

# 1 Introduction

To understand how the increasing carbon dioxide ($CO_2$) levels in the atmosphere are will change our climate, we need to know the sources and sinks of $CO_2$ separately. The main sources are fossil fuel combustion and land-use change and the main sinks are the net uptake by the terrestrial biosphere and the oceans (Friedlingstein et al., 2022). The net terrestrial biospheric sink (Net Ecosystem Exchange, NEE) results from many fluxes of which the two largest are typically Gross Primary Production (GPP) and the Total Ecosystem Respiration (TER). Knowing these gross fluxes separately will allow better estimates of the changing behaviour of the biosphere carbon sink, as GPP and TER respond differently to climate change and increasing atmospheric $CO_2$ levels (Cox et al., 2013; Ballantyne et al., 2012).

Using tracers in addition to $CO_2$ allows us to gain further insights into GPP and TER, without relying on a temperature-based function to parameterize TER as is used for Eddy Covariance (EC) measurements (e.g. Reichstein et al. (2005)). Tracers such as atmospheric $O_2$ (Keeling and Manning, 2014), and also COS, $\delta^{13}C$ or $\Delta^{17}O$ have the important advantage of sharing a process or pathway with $CO_2$ directly (Wehr et al., 2016; Whelan et al., 2018; Peters et al., 2018; Koren et al., 2019; Kooijmans et al., 2021). This allows one to use numerical models to test formulations of processes, such as stomatal and mesophyl exchange, photosynthesis, pool-specific respiration, and even turbulent canopy exchange. Atmospheric $O_2$ is directly coupled to $CO_2$ in several processes through the so-called Exchange Ratio (ER) (Keeling and Manning, 2014; Manning and Keeling, 2006; Keeling et al., 1993). This ER indicates the number of moles of $O_2$ that are consumed per moles of $CO_2$ that are produced (or vice versa) and gives a process-specific signature (Keeling, 1988).

On the global scale, the $O_2$:$CO_2$ molar ratio ER has been used to derive the global oceanic $CO_2$ sink and determine the global carbon budget (Stephens et al., 1998; Rödenbeck et al., 2008; Tohjima et al., 2019). This is done by solving the atmospheric budgets of $O_2$ and $CO_2$ with the following equations:

$$\frac{dCO_2}{dt} = F - O - B \tag{1}$$

$$\frac{dO_2}{dt} = -\alpha_F F + \alpha_B B + Z_{O_2} \tag{2}$$

where F is the fossil fuel $CO_2$ emissions, O is ocean $CO_2$ uptake, B is the net terrestrial biosphere sink of $CO_2$ and $Z_{O_2}$ indicates the ocean $O_2$ outgassing. $\alpha_F$ and $\alpha_B$ indicate the global ERs for fossil fuel combustion and the net terrestrial biosphere sink respectively. In these global studies simplified global average values ares used for $\alpha_F$ and $\alpha_B$, where $\alpha_F$ is determined from the global mixture of fuels burned, which results in 1.38 [mol/mol] (Keeling and Manning, 2014) and $\alpha_B$ was determined by laboratory measurements and a literature study of different plant and soil materials, which resulted in 1.1 [mol/mol] (Severinghaus, 1995). Furthermore, $\alpha_B$ is used to combine $O_2$ and $CO_2$ into Atmospheric Potential Oxygen (APO) (Stephens et al., 1998) which is used in determining the ocean carbon sink, and recently has also been shown to be a suitable tracer to detect fossil fuel emission reductions during the COVID-19 pandemic (Pickers et al., 2022). For these larger scale applications using

APO it is important to have good estimates for the terrestrial biosphere ERs.

On local/ecosystem scales previous studies have shown that this terrestrial biosphere ER is not a constant value of 1.1 as used on the global scale, and that is shows a certain degree of temporal and spatial variability. These studies either measured the Oxidative Ratios (ORs) from elemental composition analysis (Worrall et al., 2013; Randerson et al., 2006; Gallagher et al., 2017), or derived the ER from atmospheric concentrations measurements (Battle et al., 2019; Seibt et al., 2004; van der Laan et al., 2014). Note that there is a distinction in the terminology between ER and OR. The OR indicates the stoichiometry of specific materials, whereas the ER indicates the exchange between the atmosphere and organisms or ecosystems. By using elemental composition analysis, the OR reflects the relationship between $O_2$ and $CO_2$ over a longer time scale, of years or decades and only reflects the OR from the materials that are sampled. By using atmospheric concentration measurements for the ER, the ER reflects a shorter time scale compared to the OR, of hourly to daily time periods and it also reflects a different spatial scale, as the ER includes all processes that are originating from the footprint. The spatial scale that is covered by the ER signal depends on the type of measurements or modelling, i.e. leaf, canopy or ecosystem. Both the OR and the ER based studies showed that $O_2$:$CO_2$ molar ratio of the biosphere changes per ecosystem and over different time periods. The ER from the gas exchange experiments can furthermore be used for the separation of GPP and TER, using a specific ecosystem ER, which are determined with two alternative approaches (see Fig. 1) (Seibt et al., 2004; Stephens et al., 2007; Ishidoya et al., 2013, 2015; Battle et al., 2019). The first is the ER of the atmosphere ($ER_{atmos}$), which is the ratio of the evolution of the atmospheric $O_2$ and $CO_2$ concentration measurements over time, and the second is the ER of the forest ($ER_{forest}$), which is the ratio of the net surface fluxes of $O_2$ and $CO_2$ above the canopy, including all processes occurring below the canopy, including both vegetation and soil exchange. First attempts to estimate $ER_{forest}$ were made using one-box models (Seibt et al., 2004; Ishidoya et al., 2013). More accurate estimates of $ER_{forest}$ would be based on in situ measured $O_2$ and $CO_2$ surface fluxes, however $O_2$ currently cannot yet be measured accurately using EC techniques. Ishidoya et al. (2015) showed the first surface fluxes of $O_2$ using vertical gradients of $O_2$, an alternative technique to EC, and $CO_2$ measurements at two heights above the canopy in the surface layer in a temperate forest in Japan. Their results showed that the $ER_{forest}$ signal could be used to separate the NEE signal into GPP and TER, consistent with the separation method for EC measurements using an empirical function of air temperature.

When using $O_2$ to separately estimate GPP and TER fluxes, it is important to use the value for ER that represents ecosystem exchange. Seibt et al. (2004) showed that the signal of $ER_{atmos}$ cannot be directly linked to the exchange of carbon in the terrestrial biosphere, because in addition to the biosphere, $ER_{atmos}$ is also affected by advection, boundary layer dynamics and entrainment (Fig. 1). In contrast, Ishidoya et al. (2015) found similar values for $ER_{atmos}$ and $ER_{forest}$. So far, there is no clear consensus on which signal should be used to indicate the ER of the ecosystem. Furthermore, since atmospheric $O_2$ measurements are challenging to make, only a few studies exist that measured atmospheric $O_2$ from flasks (Seibt et al., 2004) or continuously (Ishidoya et al., 2015; Stephens et al., 2007; Battle et al., 2019) above an ecosystem and derive ER signals. The uncertainty and spatial and temporal variability of $O_2$:$CO_2$ molar ratio of the biosphere are therefore not well known

(Manning and Keeling, 2006; Keeling and Manning, 2014), and knowledge about the difference between $ER_{forest}$ and $ER_{atmos}$, its variability across difference regions and ecosystems, and how $ER_{forest}$ can be used on both the local and global scale to advance our understanding of the carbon cycle, is still limited. Therefore, more and longer in situ time series of atmospheric

90    $O_2$ measurements are needed and further understanding of $O_2$ and $CO_2$ exchange above and below the canopy is crucial to continue the pioneering work of Seibt et al. (2004), Stephens et al. (2007), Ishidoya et al. (2015) and Battle et al. (2019) and improve the application of the global biosphere ER, resulting in a better understanding of the carbon balance on local, regional and global scales.

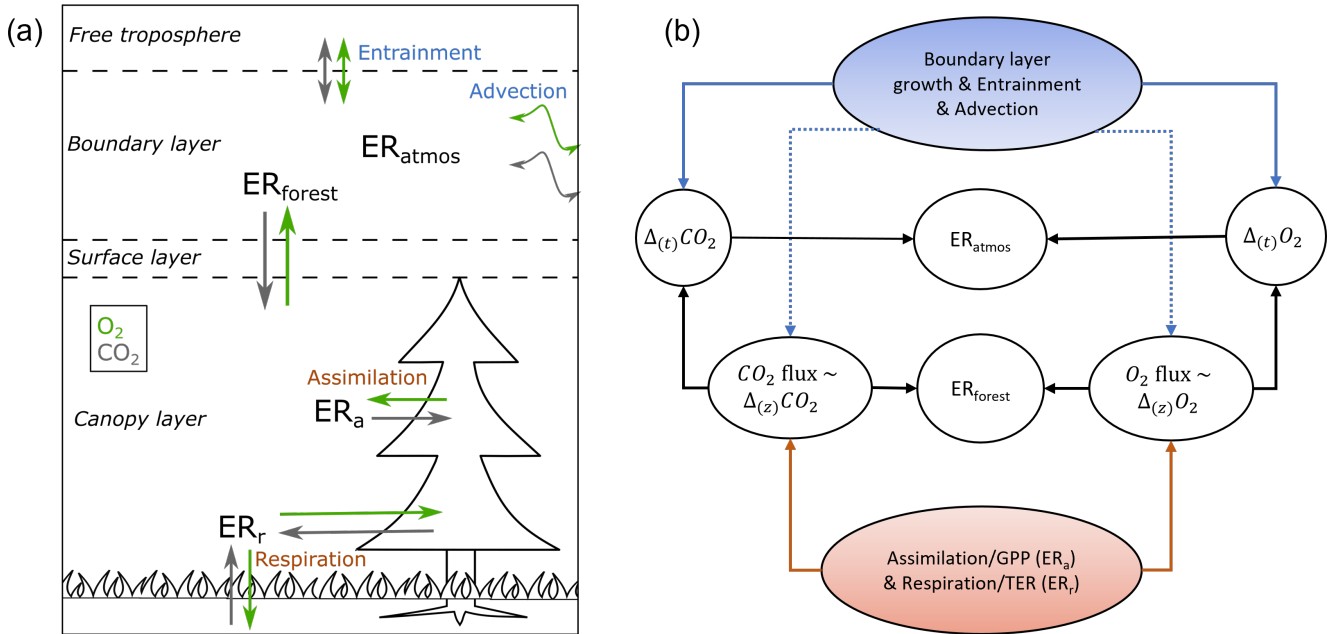

**Figure 1.** Schematic overview of the different $O_2$:$CO_2$ exchange ratio signals (ER), measured and analyzed in and above a forest, influenced by the different $O_2$ and $CO_2$ fluxes and meteorological processes (a), together with a more detailed look on which processes influence the different ER signals (b). (a) shows the direction of the surface fluxes during the day in the surface layer, which includes the roughness sublayer and the inertial sublayer. During the night the direction of the $O_2$ and the $CO_2$ surface fluxes are the other way around. The ER of the atmosphere ($ER_{atmos}$) is determined from the change over time ($\Delta_{(t)}$) in the $O_2$ and $CO_2$ concentration measurements, and the ER of the forest ($ER_{forest}$) is calculated from the surface fluxes of $O_2$ and $CO_2$ which are inferred from ($\sim$) the vertical gradient ($\Delta_{(z)}$). $ER_a$ represents assimilation processes that influence the Gross Primary Production (GPP) flux and $ER_r$ represents respiration processes that influence the Total Ecosystem Respiration (TER) flux. (b) shows the connections between the processes, measurements, and the ERs. Dotted lines indicates smaller influences of the processes that are connected to it compared to solid lines.

95    The aim of this study is to improve upon existing methods to calculate $ER_{forest}$ and get a better comparison of the $ER_{atmos}$ and $ER_{forest}$ signals. We carried out a measurement campaign in Hyytiälä, Finland, for two short periods in spring/summer

2018 and 2019 where both $O_2$ and $CO_2$ were measured at two heights with a setup including a differential fuel cell analyser for $O_2$. We used our measurements to determine the diurnal behaviour of the relation between the concentrations and the fluxes of $O_2$ and $CO_2$, by using either one or both measurement heights on the tower. The objectives of this study are: 1) to extend the existing continuous $O_2$ records, 2) to calculate the $O_2$ surface fluxes in a boreal forest for the first time, 3) to combine the $O_2$ and the $CO_2$ fluxes, to calculate $ER_{forest}$ from these fluxes, and to compare the $ER_{atmos}$ and $ER_{forest}$ signals, and 4) use $ER_{forest}$ to estimate GPP and TER fluxes.

In this paper, we first describe the measurement site, experimental setup and methods used to derive $O_2$ fluxes and the different ER signals (Sect. 2). We present the measurements for the whole campaign and select a representative day to determine the most suitable approach for deriving $O_2$ fluxes and to determine $ER_{forest}$ (Sect. 3). A detailed evaluation and discussion of our $ER_{atmos}$ and $ER_{forest}$ signals is given in Sect. 4. We finalize with our conclusion about the diurnal variability of the ER signals for a representative day of a boreal forest (Sect. 5).

## 2 Methods

To determine $ER_{atmos}$ and $ER_{forest}$, and its diurnal variability, we measured $O_2$ and $CO_2$ continuously at two heights above a boreal forest during two short campaigns at Hyytiälä. These 'OXHYYGEN' (OXYGEN at HYYtiälä) campaigns took place in the spring/summer of 2018 (03-Jun through 02-Aug) and 2019 (10-Jun through 17-Jul). In this section, we describe the measurement site and instrumental setup, as well as the methods used to determine the $O_2$ and $CO_2$ fluxes from the measured vertical gradient and the ER signals.

### 2.1 Measurement site

The measurements were made at Hyytiälä SMEAR II Forestry Station of the University of Helsinki in Finland (61° 51'N, 24°17' E, +181 MSL); this site is described in more detail in e.g. Hari et al. (2013). The SMEAR II station is a boreal site within the European Integrated Carbon Observation System (ICOS) network with atmospheric and ecosystem measurements. The SMEAR II station is located inside a homogeneous forest of Scots pine trees (*Pinus Sylvestris*) with a dominant canopy height of 18 m and some silver birch and aspen trees. The forest floor is covered with mosses and herbs. The soils are podzols on top of glacial till. A large lake is located close (around 550 m) to the measurement site and has a fetch of 250 m over the dominant wind direction of 230°. The footprint of the site is mostly influenced by natural sources, with the atmospheric signal dominated by forest exchange (Carbon Portal ICOS RI, 2022). The measurement site includes several towers, including a 128m tall tower and a 23 m high walk up tower, where atmospheric variables and gas concentrations are continuously measured. The operational data from this tower are publicly available online at http://avaa.tdata.fi/web/smart/smear/. Our $O_2$ and $CO_2$ measurement setup was installed in a cabin at the bottom of the 23 m high tower, and air was sampled from aspirated inlets (Blaine et al., 2006), installed at 23 m in the smaller tower and at 125 m in the tall tower, 5m and 107m above the canopy

height respectively. We used both levels to calculate the vertical gradient for the flux calculations (Sect. 2.3).

## 2.2 Experimental setup

The measurement setup is based on the instrument used in van Leeuwen and Meijer (2015), following the methods in van der Laan-Luijkx et al. (2010) and Stephens et al. (2007). $O_2$ is measured with a Sable Systems "Oxzilla II" fuel cell based instrument and $CO_2$ is measured with an ABB continuous gas analyzer "URAS26", which is a non-dispersive infrared (NDIR) photometer. The gas handling schematic is shown in Fig. 2.

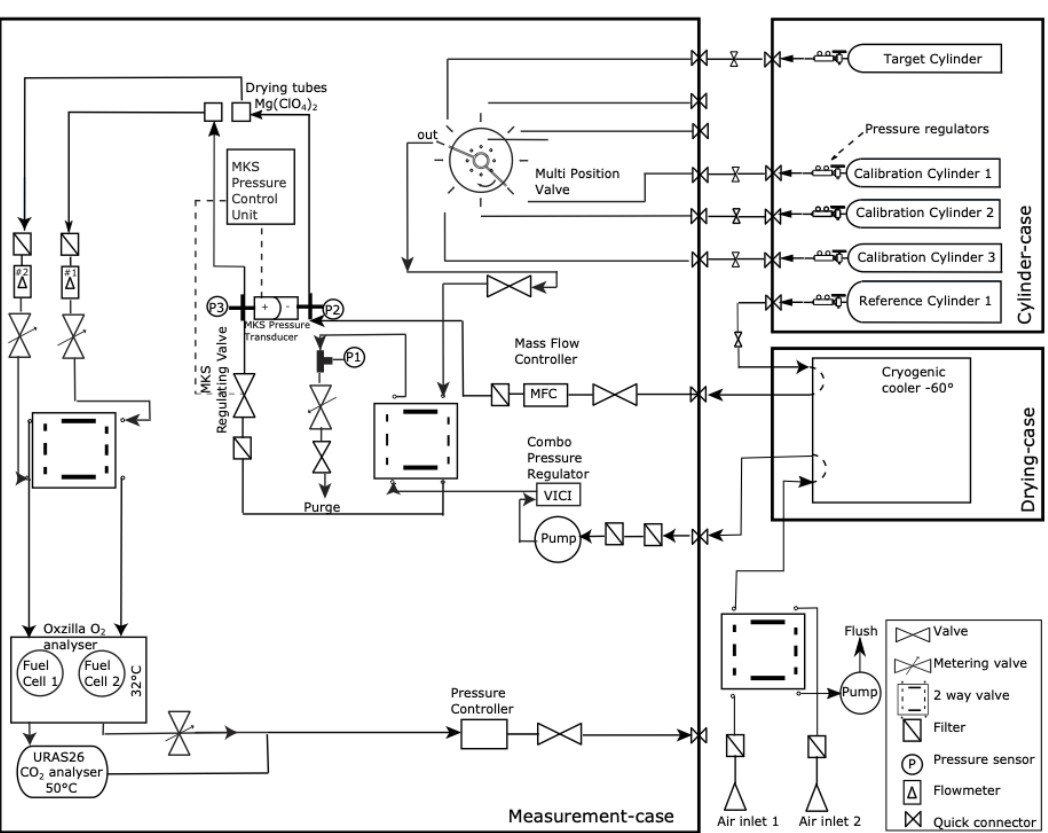

**Figure 2.** Schematic overview of the measurement setup used at Hyytiälä. The setup includes an Oxzilla $O_2$ fuel cell analyser and a URAS26 NDIR $CO_2$ analyser. The system measured air sampled from two heights of either 23 or 125 meters.

Air was pumped from either 23 or 125 metre height to the measurement system at the base of the tower. Both inlet lines were continuously flushed, where either one of the heights is measured by the system with a sample flow of around 120 ml/min and the other flushed to the room with a higher flow rate of around 2 litre per minute, which allows fast switching between the two heights. We switched between the inlets every half hour to match the EC measurements of ICOS that were already present in the tower and to get a more stable signal of $O_2$. The air of the selected inlet was first cooled to -60 °C with a cryogenic cooler to remove water vapour from the air, before entering the system. Second stage drying of the air streams was done with magnesium perchlorate $(Mg(ClO_4)_2)$ traps. The sample air was continuously measured against a reference gas (differentially for $O_2$, and alternatively for $CO_2$), and the pressure in both sample and reference line were matched to be the same using a pressure control system (MKS Instruments, types 223B, 248A and 250E for the pressure transducers, regulating valve and control system respectively). The reference and sample lines were switched every 2 minutes between the two fuel cells in the Oxzilla analyser. We measured a set of 3 calibration cylinders and 1 target cylinder every 23 hours for half an hour per cylinder.

The measurements of these calibration gases allowed calibration of our measurements against the international Scripps Institution of Oceanography (SIO) scale for $\delta O_2/N_2$. We did that by using cylinders that are filled in the laboratory at the University of Groningen, where they were calibrated with the primary Scripps cylinders (Nguyen et al., 2022). The $O_2$ measurements are normally expressed as $\delta O_2/N_2$ ratios in 'per meg' units instead of mole fraction (ppm), since $O_2$ is not a trace gas because of its high abundance of 20.95%, and therefore the mole fraction varies due to changes of other gases, such as $CO_2$ (Keeling et al., 1998). $\delta O_2/N_2$ is defined as:

$$\delta(O_2/N_2) = \left( \frac{(O_2/N_2)_{\text{sample}}}{(O_2/N_2)_{\text{reference}}} - 1 \right) \cdot 10^6 \text{ [per meg]} \tag{3}$$

For simplicity, in this paper we use the term $O_2$ instead of $\delta O_2/N_2$, and we use the term 'concentration' rather than 'mole fraction' when discussing both $CO_2$ and $O_2$. Equation (3) indicates a change compared to a reference level. Negative values therefore indicate concentrations of $O_2$ lower than the reference value. To allow comparison of changes in $CO_2$ and $O_2$ directly, we converted the units of $O_2$ from per meg to ppm equivalents (ppmEq), where a change of 1 ppm $CO_2$ corresponds to a 4.77 per meg change in $O_2$ (Tohjima et al., 2005; Kozlova and Manning, 2009).

We modified the method described in van der Laan-Luijkx et al. (2010), to calibrate the measurements. The raw $CO_2$ measurements have a frequency of one measurement per six second, the raw $O_2$ measurements have a frequency of one measurement per second and both give 1 value every 4 minutes in the form of $\Delta CO_2$ and $\Delta(\Delta)O_2$ respectively. $CO_2$ is measured on a single cell instrument, and therefore $\Delta CO_2$ is the difference between the 2-minute averages of the sample air (S) and the reference cylinder (R), giving (S-R). For the 2-minute averaged $CO_2$ measurements, the last 78 seconds of each 2 minute period are used. Note that for $CO_2$, the NDIR system is different compared to other systems used and therefore does not need a zero-gas (Pickers et al., 2017). $O_2$ is measured on a double cell instrument, and therefore gives a double differential signal. The $\Delta(\Delta)O_2$ is the difference between the 2-minute averaged difference between S and R and the 2-minute averaged difference between R and S ((S-R)-(R-S)). For the 2-minute averaged $O_2$ values, the last 100 seconds of each 2 minute period are used. In

2019, the MKS pressure control valve was not functioning optimally, and led to a small instability in the differential pressure between the sample and reference lines. We therefore corrected the 4-minute values of $\Delta(\Delta)O_2$ for this deviation measured by the MKS differential pressure sensor (PMKS), by multiplying $\Delta(\Delta)O_2$ with 0.095*PMKS, which we derived based on the measurements of the calibration cylinders. In 2018, there was no instability in the pressure control valve, and therefore no correction was applied in that year. The PMKS deviations correlated with temperature and increased towards the end of the 2019 campaign. Figure B1 in Appendix B shows that the highest corrections were made during the mid-day at the end of the campaign. The $O_2$ vertical gradient is hardly affected by the correction as it is the difference between measurements at two heights which are both undergoing the same deviation.

For both $CO_2$ and $O_2$, the 4-minute values were subsequently used to calculate half hourly means, where we excluded the first 4-minute value after the heights are switched, together with the measurements that did not fall inside the boundary based on the median absolute deviation (MAD) (Rousseeuw and Verboven, 2002). For every half hourly mean, a standard error is calculated (see Eq. (4)) which is used in further analysis to determine the uncertainty of our measurements.

The linear calibration response functions for both $O_2$ and $CO_2$ were calculated for every measurement period of the calibration cylinders, which was about every 23 hours. For the response functions, we used a constant slope based on the mean of all the calibration slopes measured in the specific year. The y-intercept of the response functions were interpolated to the time of the measurement, based on the two calibrations bracketing the measurement time. To facilitate the comparison of the $O_2$ and $CO_2$ measurements of the two heights and allow flux calculations based on the vertical gradient, we interpolated the data to one measurement for every 30 minutes for each height. Based on the target cylinders, measured during the calibration period, the stability of the long-term measurements were determined (Table 1). A different target cylinder, with different composition of air for 2019 compared to 2018 was used, which resulted in different outcomes for the standard deviation (std) and the mean difference for these periods. The mean difference is calculated from the target measurements at Hyytiälä compared to the calibrated values using the SIO cylinders in Groningen. The measurement period of 2018 was also longer and therefore more points were included for the std and mean difference calculations. The long-term measurement precision of this device throughout the duration of the two measurement campaigns compared to the recommendations of the World Meteorological Organisation (WMO) will be further discussed in Sect. 4.1.

**Table 1.** The mean difference and the standard deviation (std) of the target cylinder measurements of $O_2$ and $CO_2$ for the 2018 and 2019 periods separately, together with the number of data points used to calculate these specific values.

| | **2018** (03-06 through 01-08) | | | **2019** (16-06 through 17-07) | | |
|---|---|---|---|---|---|---|
| | Std | Mean difference | Number of points | Std | Mean difference | Number of points |
| $O_2$ [per meg] | 16 | 28 | 53 | 19 | 22 | 22 |
| $CO_2$ [ppm] | 0.07 | 0.7 | 53 | 0.07 | 0.5 | 22 |

## 2.3 Data analysis

For the analyses presented in this paper we needed representative diurnal cycles of $O_2$ and $CO_2$. We looked for a representative day in 2019 where little to no clouds were present, no unexpected behaviour in the diurnal cycles for potential temperature, specific humidity and $CO_2$ occurred, (for example caused by advection) and where the $O_2$ data showed a clear difference between the two measurement heights. We used data from 2019 instead of 2018 because 2018 saw a large-scale drought in Europe, and 2019 was less extreme and closer to a typical boreal summer (Peters et al., 2020). However, no single representative day could be found in our 2019 record, where the $O_2$ data showed a clear negative vertical gradient during the day and positive during the night, in combination with the above-mentioned meteorological criteria. We therefore choose a sequence of days to create an aggregate day based on the average of several days, which is representative for this time of the year in Hyytiälä, following the same method used by Ishidoya et al. (2015). The main criterion was that the vertical $O_2$ gradient had to be negative during the day and the negative relationship between the change of $O_2$ and $CO_2$ concentrations over time at 23 m was present during the entire day. This resulted in selecting the period of 7 through 12 July 2019 to create the representative day which we used in all subsequent analyses. The half-hourly values for the representative day are the averages of the data points of the individual half hourly values for each day in the selected period. Each time step has an uncertainty which is based on the error propagation of the standard error (SE) of the 30-minute averages for each day in the aggregate and is calculated for each time step with:

$$SE_{aggr} = \frac{\sqrt{\sum SE_{day}^2}}{n} \qquad (4)$$

Where n is the number of days included in the aggregate, $SE_{day}$ is the Standard Error of the 30-minute average of each individual day and the $SE_{aggr}$ is the resulting Standard Error of a 30-minute value for the representative aggregate day.

For the representative day, the two $O_2 : CO_2$ Exchange Ratio (ER) signals, $ER_{atmos}$ and $ER_{forest}$, were determined. $ER_{atmos}$ is concentration-based and is expressed as:

$$ER_{atmos} = -\frac{\Delta_{(t)}O_2}{\Delta_{(t)}CO_2} \qquad (5)$$

Where both $\Delta_{(t)}O_2$ and $\Delta_{(t)}CO_2$ are the change in concentration over a selected time period (t). This is a unit-less quantity as it represents mol $O_2$ per mol $CO_2$. $ER_{atmos}$ was determined by the slope of a linear regression between the concentration of $O_2$ and $CO_2$ at the same height over a specific time period (Seibt et al., 2004; Stephens et al., 2007; Ishidoya et al., 2013; Battle et al., 2019). The selected time periods were based on the period when $O_2$ and $CO_2$ had the highest negative correlation. Throughout the day, this could be divided into three periods when different processes dominate (Fig. 1). It starts with the period during the night where the atmosphere is stable and respiration becomes the dominant surface flux (P1), and therefore the $CO_2$ concentration increases and the $O_2$ concentration decreases. Subsequently, when the sun starts to rise, the boundary layer height starts to grow and entrainment of air from the free troposphere influences the surface measurements (P2) (Vilà-Guerau de Arellano et al., 2004). Here the $CO_2$ concentration decreases rapidly and the $O_2$ concentration increases rapidly. Finally, the

period starts when the effect of boundary layer dynamics and entrainment decreases and the assimilation flux dominates (P3), here the $CO_2$ concentration decreases less rapidly and the $O_2$ concentration increases less rapidly. We calculated a $ER_{atmos}$ signal with Eq. (5), for the night-time (P1), the day-time (by either focusing on only P3 or both P2 and P3) and the complete day (P1 + P2 + P3). The exact boundaries of these periods have to be estimated. To be certain about the exact times that should be taken as the boundaries for each period, an atmospheric model is needed.

$ER_{forest}$ is flux-based and is expressed as:

$$ER_{forest} = -\frac{F_{O_2}}{F_{CO_2}} \tag{6}$$

Where both $F_{O_2}$ and $F_{CO_2}$ are the net mean turbulent surface fluxes above the canopy of $O_2$ and $CO_2$ over a selected time period (Seibt et al., 2004; Ishidoya et al., 2015). We derive the fluxes of $O_2$ and $CO_2$ using the vertical gradient (see next paragraph and Eq. (7)). The selected time periods for $ER_{forest}$ were chosen such that the transition periods between the nighttime with a stable atmosphere (when the respiration flux dominates) to the daytime with a well mixed atmosphere (when assimilation dominates), were excluded. By excluding the transition periods, we removed the periods where the gradients of both $CO_2$ and $O_2$ were close to zero. This was done because a very small gradient makes it difficult to calculate a flux and therefore the $ER_{forest}$, and also because during this period entrainment is the most dominant process. The exact duration of the transition periods was based on the maximum and minimum of both the friction velocity and the height of 27 m (z) divided by the Monin Obukov Length (L). The friction velocity and (z/L) indicate the measure of turbulence of the atmosphere (Stull, 1988). The mean of the remaining data points of the $CO_2$ and $O_2$ flux during the stable atmosphere period was used to calculate the $ER_{forest}$ signal of the night and the mean of the remaining data points of the $CO_2$ and $O_2$ flux during the mixed atmosphere period was used to calculate the $ER_{forest}$ signal of the day. The $ER_{forest}$ for the entire day is taken as the average $CO_2$ and $O_2$ flux over the entire day. For this average no periods are excluded and all the data points over the 24 hours are taken into account. Taking the average daily fluxes to derive $ER_{forest}$ is a slightly different approach compared to the study of Ishidoya et al. (2015), who use the regression line between $\Delta_{(z)}O_2$ and $\Delta_{(z)}CO_2$ to determine $ER_{forest}$.

Currently, unlike for $CO_2$, the $O_2$ flux cannot be measured directly with an EC system. Instead, the flux can be inferred from the flux-gradient method. To calculate the flux of a certain scalar ($\phi$) with the flux-gradient method, the following equation was used (Stull, 1988):

$$F_\phi = -K_\phi \cdot \frac{\partial \overline{\phi}}{\partial z} \tag{7}$$

Where $F_\phi$ is the surface flux of $\phi$, K is the exchange coefficient and ($\partial \overline{\phi}/\partial z$) is the vertical gradient of $\overline{\phi}$. To determine the $O_2$ flux with Eq. (7) ($\overline{\phi} = \overline{O_2}$), the exchange coefficient of $O_2$ ($K_{O_2}$) needs to be determined. Ishidoya et al. (2015) assumed that $K_{O_2} = K_{CO_2}$ and determined $K_{CO_2}$ by dividing the $CO_2$ flux, measured with EC, by the $CO_2$ vertical gradient between two measurement levels. However, the exchange coefficient can also be determined with other methods that for example only need two measurement heights for the vertical gradient. In this study, we explore these different options for calculating $K_{O_2}$.

The EC measurements of the $CO_2$ flux were used as a reference, to determine the most suitable approach. The most suitable approach to infer the $O_2$ flux is then used for both $K_{CO_2}$ and $K_{O_2}$. During this study we derive the surface flux in the surface layer (Fig. 1) and we assume that the surface flux stays constant in this surface layer, which consists of the roughness sublayer and the inertial sublayer.

We categorized the methods to determine the most suitable K into two groups: The observation-based approach (also called the K-theory (Stull, 1988) or the modified Bowen ratio method (Meyers et al., 1996)) and the theoretical approach (following the similarity theory (Dyer, 1974)). For the observation-based methods, we determined the exchange coefficient (K) in Eq. (7) by dividing a flux measured at 27 m, using an EC system, by a 3-height (16 m, 67 m and 125 m) vertical gradient of a specific scalar. Ishidoya et al. (2015) used this approach to calculate their $O_2$ flux, using the $CO_2$ flux and vertical gradient of two levels. Next to $CO_2$, we also calculated K using potential temperature ($\theta$) for the observation-based approach.

For the theoretical approach, the K in Eq. (7) is determined with the Monin-Obukov Similarity Theory (MOST) (Dyer, 1974), where logarithmic surface layer scaling applies for K and empirical similarity functions are used to describe the effect of atmospheric stability. In addition, we used a correction which takes into account the effect of the roughness sublayer (see Appendix for details). The SMEAR II data at 27 meter were used for the calculations with MOST. When only two heights for the gradient calculations are available, there is an option to integrate Eq. (7) (de Ridder, 2010). We tested both the application with and without integration in this study. We used the ICOS data, available at the SMEAR II station, for the K calculations. For the $CO_2$ EC measurements, we used the gap-filled data to correct for the storage below the measurement height of the EC. Gap-filling was applied when the friction velocity (u*) was below 0.4 m s$^{-1}$ (Kulmala et al., 2019). The Appendix gives a more elaborate explanation and provides equations of the different methods used to determine the exchange coefficients used in this study.

Finally, we select the $K_\phi$ from either the observation-based or the theoretical approach that produced $CO_2$ flux results from our $CO_2$ vertical gradient measurements that showed the best comparison to the EC $CO_2$ flux measurements. This K was used to calculate the $O_2$ and $CO_2$ fluxes, together with the vertical gradient from measurements collected during our campaigns. For our campaigns, we only have $O_2$ and $CO_2$ measurements at two heights (23 m and 125 m), which means that ($\partial \overline{\phi}/\partial z$) changes into ($\Delta \overline{\phi}/\Delta z$) and the gradient was calculated with finite differences.

After both the $CO_2$ and $O_2$ fluxes were determined, resulting in $ER_{\text{forest}}$, we subsequently calculated the $O_2 : CO_2$ exchange ratio signals for the assimilation processes ($ER_a$) and the respiration of the ecosystem ($ER_r$) with the following equations (Seibt et al., 2004; Ishidoya et al., 2015):

$$NEE = -GPP + TER \tag{8}$$

$$NEE \cdot ER_{\text{forest}} = -GPP \cdot ER_a + TER \cdot ER_r \tag{9}$$

Where the NEE is the Net Ecosystem Exchange, GPP is the Gross Primary Production and TER is the Total Ecosystem Respiration. GPP and TER are always positive by definition, representing uptake and release by the ecosystem respectively. Therefore, when GPP is larger than TER the resulting negative NEE values represent carbon uptake by the ecosystem. First, we assumed that nighttime NEE is equal to TER, which meant that the nighttime $ER_{forest}$ signal is equal to $ER_r$. We assumed that the processes that contributed to the $ER_r$ keep the same ratio between $O_2$ and $CO_2$ during the entire day and therefore we used a constant $ER_r$ for the entire day. We base this assumption on studies that showed that the variability of $ER_r$ highly depends on the bulk soil respiration (Hilman et al., 2022; Angert et al., 2015). No large changes occur in the soil temperature and the soil moisture during our (representative) diurnal cycle, and therefore we assume that the $ER_r$ of the bulk soil respiration stays relatively constant and with that the $ER_r$ of the ecosystem also stays constant over the entire day. Subsequently, we calculated $ER_a$, for both the entire diurnal cycle and the daytime using Eq. (9) with the corresponding $ER_{forest}$ and the constant $ER_r$. We used ICOS NEE EC measurements from the SMEAR II station at a level of 27 meters in the 128 m high tower. The GPP fluxes at Hyytiälä are calculated with either of the following 2 approaches: 1) when NEE EC measurements are available, GPP is calculated as the difference between the NEE EC measurements and the respiration flux, which is calculated using a temperature function or 2) when NEE EC measurements are not available, GPP is calculated using an equation that is based on the air temperature and light (Photosynthetically active radiation (PAR)). A more detailed description of these calculations is given by Kulmala et al. (2019) and Kohonen et al. (2022).

By estimating $ER_r$ and $ER_a$ of this boreal forest, we created the opportunity to apply atmospheric $O_2$ measurements to separate NEE into GPP and TER (the $O_2$ method). We calculated $ER_r$ and $ER_a$ for the representative day using Eq. (8) and (9), and use these to calculate GPP and TER for another representative day. We selected 13 through 15 June to create a new second aggregate day and to calculate a new $ER_{forest}$ signal for the entire day (see Fig. 3 d and e) for a detailed view on the measurements of those days). These three days were chosen because in 2019 they showed the clearest diurnal cycle of $O_2$ and a negative $O_2$ gradient, aside from 7 through 12 July, used above. We assume here that the $ER_r$ and $ER_a$ calculated for the period from 7 through 12 July are representative for the period from 13 through 15 June. Studies show that the $ER_r$ (Hilman et al., 2022) and $ER_a$ (Bloom, 2015; Fischer et al., 2015) vary with changing soil and atmospheric conditions. The periods for both representative days are relatively close in time and therefore have similar conditions in the soil and the atmosphere, and we can therefore assume that the $ER_r$ and $ER_a$ based on the 7 through 12 July data can also be applied to the 13 through 15 June period. By using the $ER_r$ and $ER_a$ determined for the first representative day (7-12 July), and $ER_{forest}$ and NEE for the second representative day (13-15 June), we calculated GPP and TER from NEE for this second representative day. By comparing the GPP and TER fluxes of the $O_2$ method to the GPP and TER fluxes of the temperature-based function of ICOS (EC method), we could demonstrate how accurate the $O_2$ method is (Sect. 3.4). Both Seibt et al. (2004) and Ishidoya et al. (2015) also applied the $O_2$ method, however both these studies used chamber measurements to first determine $ER_a$ and $ER_r$ and then used Eq. (8) and (9) to infer GPP and TER. Unfortunately we did not have chamber measurements of both $O_2$ and $CO_2$ available and therefore we used Eq. (8) and (9) to calculate $ER_a$ and $ER_r$. This means that these two equations can be used in two ways: to determine

the $ER_a$ and $ER_r$ signal, or to separate NEE into GPP and TER.

The footprint of the calculated $O_2$ and $CO_2$ surface fluxes which also represents the footprint of the $ER_{forest}$, $ER_r$ and $ER_a$ signals for the representative aggregate day is shown in Fig. B2 in Appendix B. The footprint is based on the method of Kljun et al. (2015) where for the height, the geometric mean between 125 m and 23 m is used. The footprint analysis shows that
the surface fluxes are mainly influenced by the forest surrounding the tower and that the lake located close to the tower is not influencing the signal. The footprint of the $O_2$ and $CO_2$ concentrations and therefore the footprint of the $ER_{atmos}$ signal can be found in the document of (Carbon Portal ICOS RI, 2022). This concentration footprint analysis shows that with an average wind direction of north to northeast between 7 through 12 July 2019, the concentrations measured are mainly originated from forest exchange, with hardly any influence of urban sources.

Table B1 in Appendix B gives a complete overview of which data are used for each part of this research for the two different aggregate days.

## 3   Results

### 3.1   $O_2$ and $CO_2$ time series

The calibrated half hourly measurements of $O_2$ and $CO_2$ for 2018 and 2019 are shown in Fig. 3, together with the vertical gradients between the two measurement heights. The $O_2$ measurements are shown here converted from per meg to ppmEq, to allow comparison of the diurnal variability for $CO_2$, and to calculate the ER signals. The differences between the 23 m and 125 m measurements are observable for both $CO_2$ and $O_2$. During both campaigns in 2018 and 2019, the diurnal behaviour of
the $O_2$ concentrations are anticorrelated with the $CO_2$ concentrations. This anticorrelation between $O_2$ and $CO_2$ is also visible from the gradient measurements, despite the relatively high uncertainty of the $O_2$ measurements as described in Sect. 2.2 and further elaborated on in Sect. 4.1. The period 7 through 12 July 2019 shows the most clear negative relationship between the $O_2$ gradient and the $CO_2$ gradient, and also had the most suitable meteorological conditions and was therefore selected for the aggregate representative day (Sect. 2.3). The period 13 through 15 June shows a less clear anticorrelation between the vertical
gradients of $O_2$ and $CO_2$ (Fig. 3 d and e), but with clear diurnal cycles of $O_2$ and $CO_2$ suitable for the purpose of our second aggregate day (see Sect. 3.4).

### 3.2   Diurnal cycles

The measurements of $O_2$ and $CO_2$ and their vertical gradient for the representative day, are shown in Fig. 4. There are no
measurements between 20:00 and 22:00 because the calibration cylinders were measured during this period. For 7 through 12

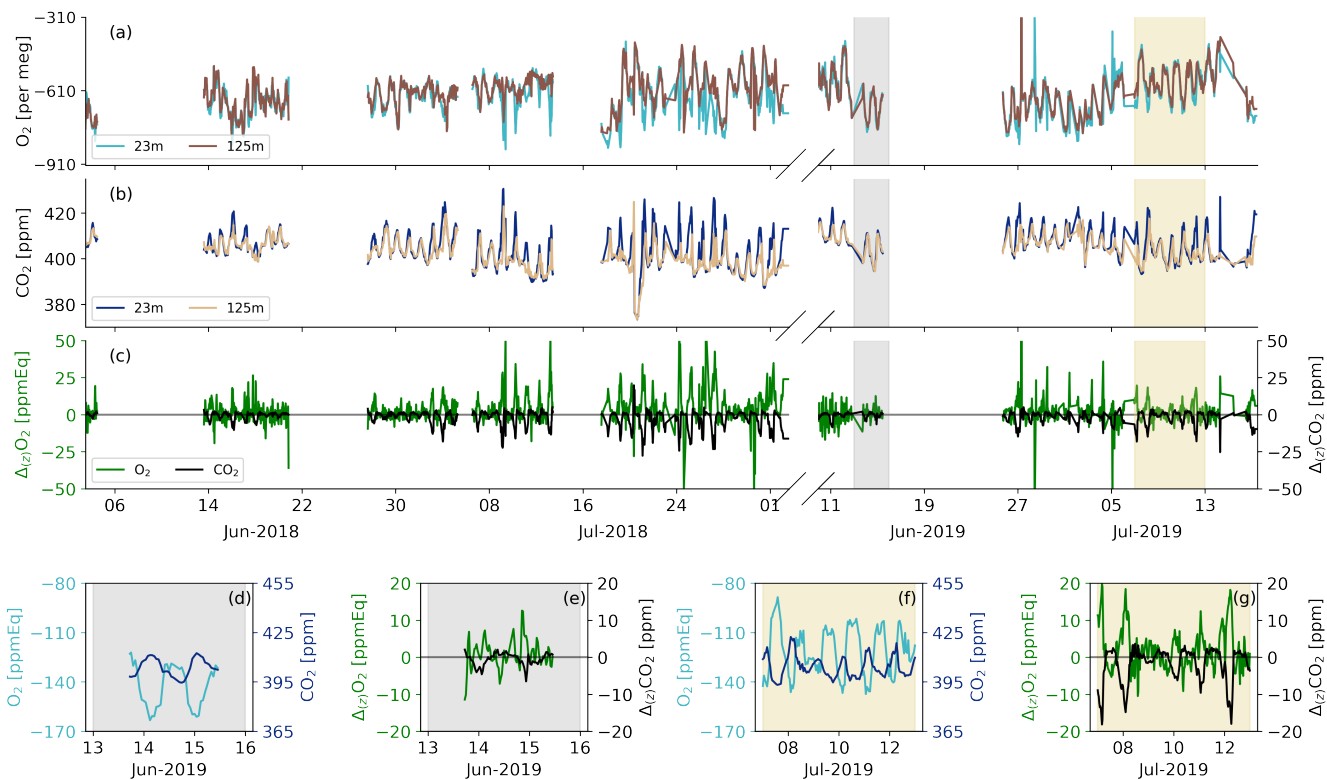

**Figure 3.** The half hourly average $O_2$ (a) and $CO_2$ (b) concentrations at Hyytiälä for spring/summer of 2018 and 2019 for the 125 m and 23 m height levels, together with the vertical gradient ($\Delta_{(z)}$) between these two heights (c) for both $O_2$ and $CO_2$. The shaded areas indicates the dates that were selected for the aggregate representative day (7 through 12 July 2019: yellow) and the second representative day to test the $O_2$ method (13 through 15 June 2019: grey). The selected days for the aggregate representative days are shown in more detail for the 23 m measurements and the gradients for 13 through 15 June (d) and (e) and for 7 through 12 July (f) and (g) for both $O_2$ and $CO_2$.

July we used a fixed calibration time, as radiosondes were launched (not shown) during this period and we wanted to make sure we captured the morning transition to compare with these radiosondes. Note that the daylight length at Hyytiälä is long at this time of the year, with sunrise at 03:00 and sunset at 22:00. We compared our $CO_2$ observations with ICOS $CO_2$ measurements at the same height, which shows that both instruments compare well overall, with a mean difference of $0.70 \pm 0.65$ ppm during
the period 7 through 12 July. The comparison between the two devices was a bit difficult because of the different timing of the measurements. The diurnal cycles of $O_2$ and $CO_2$ (Fig. 4a) clearly show anti-correlated behaviour between $CO_2$ and $O_2$, which is especially visible during nighttime (23:00 - 04:00) and the morning transition (05:00 - 13:00).

Figure 4 shows four different periods that can be linked to the periods to calculate $ER_{atmos}$, described in Sect. 2.3. P1 is
visible between 23:00-04:00, where respiration starts to dominate the signal and therefore the $O_2$ concentration decreases and the $CO_2$ concentration increases, in a decreasing boundary layer height dominated by thermal stratification. P2 becomes visible

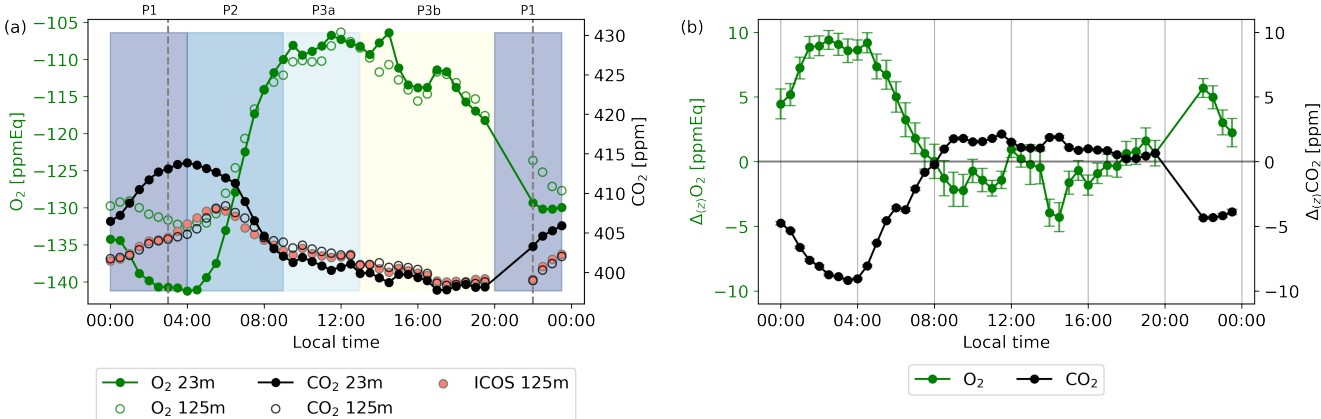

**Figure 4.** Diurnal cycles (in local wintertime, +2UTC) of the $O_2$ and $CO_2$ concentrations for the 23 m and 125 m height levels (a) and the vertical gradient between both levels with the uncertainty of both $O_2$ and $CO_2$ of the representative day, taken as the average values of 7 through 12 July 2019 (b). The $CO_2$ measurements of the ICOS setup are shown in (a) for comparison to the $CO_2$ setup measured during our campaigns. The shaded colors indicate the selected different periods where the most dominant processes are: stable atmosphere and respiration (00:00-04:00, P1), entrainment, boundary layer growth and assimilation (04:00-09:00, P2), convective conditions and assimilation (09:00-13:00, P3a), the same as P3a, plus a remaining artefact for the $O_2$ measurements after the pressure correction as explained in the text (13:00-20:00, P3b). The vertical dotted lines indicate the sunrise (3:00) and sunset (22:00). The error bars in panel (b) are half-hourly standard errors based on the error propagation of the standard errors of the data points in (a) (not shown), which were based on Eq. (4).

around 04:00 and stops around 09:00, where entrainment, the growing boundary layer and the onset of photosynthesis causes a steep increase in the $O_2$ concentration and a steep decrease in the $CO_2$ concentration. P3 can be divided into P3a and P3b and is visible between 09:00-20:00. Between 09:00-13:00 (P3a), the photosynthesis flux starts to dominate and both the $O_2$
and $CO_2$ concentration increase and decrease less rapidly. Between 13:00-20:00 (P3b) the $O_2$ concentration starts to decrease, while the assimilation flux still dominates, which is a remaining artefact from the pressure correction that we applied due to the instability of the MKS pressure transducer (see Sect. 2.2). As shown in Fig. B1 in Appendix B, higher daytime temperatures cause larger PMKS deviation and therefore the effect of the pressure correction is largest during the mid-day, leading to a larger uncertainty in the observations in that time period. The boundary of 20:00 between P3b and P1 was difficult to determine
because we missed some measurements due to the calibration period and the remaining measurements around this time have a deviation caused by the pressure transducer. Measurements at both levels show this same diurnal behaviour, however it is more pronounced closer to the vegetation (the 23 m level).

The difference between the two heights results in a vertical gradient (Fig. 4b). Similar to the diurnal cycle of the concen-
trations, the diurnal cycles of the gradients of $O_2$ and $CO_2$ also show anti-correlated behaviour. At 08:00, the $CO_2$ gradient changes from negative to positive and the $O_2$ gradient changes from positive to mostly negative, reflecting $CO_2$ being transported downwards and $O_2$ upwards respectively. The magnitude of the gradient depends on the degree of vertical mixing. The

sign of the gradients changes during the day, because the lowest level (23 m) is more directly influenced by forest carbon exchange compared to the highest level (125 m). Around the time of sunset, the $CO_2$ gradient changes from positive to negative and the $O_2$ gradient changes from negative to positive, because the lowest measurement level (23 m) is now influenced more by respiration processes of the forest and soils compared to the highest measurement level (125 m). The error bars are based on the error propagation of the standard errors of each half-hourly data point, that were calculated with Eq. (4). The gradient of $O_2$ is hardly affected by the PMKS correction (see Fig. B1), as measurements at both the heights are affected similarly.

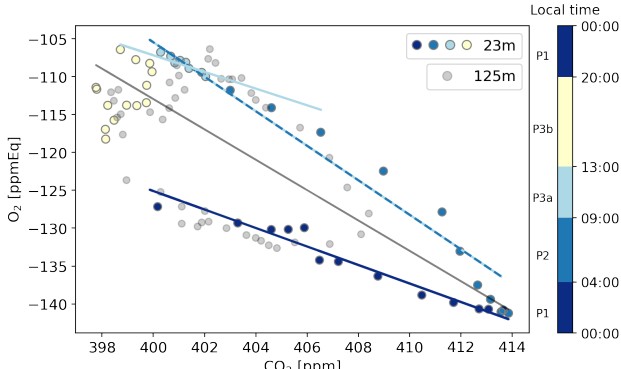

**Figure 5.** The $O_2$ concentration plotted against the $CO_2$ concentration for the representative day (in local wintertime, +2UTC), with the 23 m level in coloured points per period representing different dominant process and the 125 m level in grey points. The dominant processes are: respiration (00:00-04:00), entrainment (04:00-09:00), assimilation (09:00-13:00), a remaining artefact after the pressure correction due to the instability of the MKS pressure transducer becomes visible (13:00-20:00). The linear regression lines indicate the exchange ratio of the atmosphere ($ER_{atmos}$) during the time with a specific dominant process.

By using Eq. (5), we calculated four distinct $ER_{atmos}$ signals for different periods throughout the day at 23 m, and to a smaller degree at 125 m (Fig. 5 and Table 3). The same periods as shown in Fig. 4 are visible in Fig. 5. This results in an $ER_{atmos}$ during the night (P1) of $1.22 \pm 0.02$ and two different possibilities for the $ER_{atmos}$ signal during daytime. By combining both P2 and P3a we get a signal of $2.28 \pm 0.01$ and by focusing only on P3a, which excludes the entrainment and the boundary layer dynamics, we get a signal of $1.10 \pm 0.12$. Last, by combining all the periods (P1, P2, P3) we get a signal for the complete day of $2.05 \pm 0.03$. The uncertainties given here only represent the uncertainty of the slopes from the regression lines in Fig. 5. The high values for the $ER_{atmos}$ signal of the entire day and the daytime signal that includes entrainment and the boundary layer dynamics are not very realistic to represent an ER for the forest, and shows that we should be careful when using $ER_{atmos}$. This will be elaborated on in Sect. 4.2.

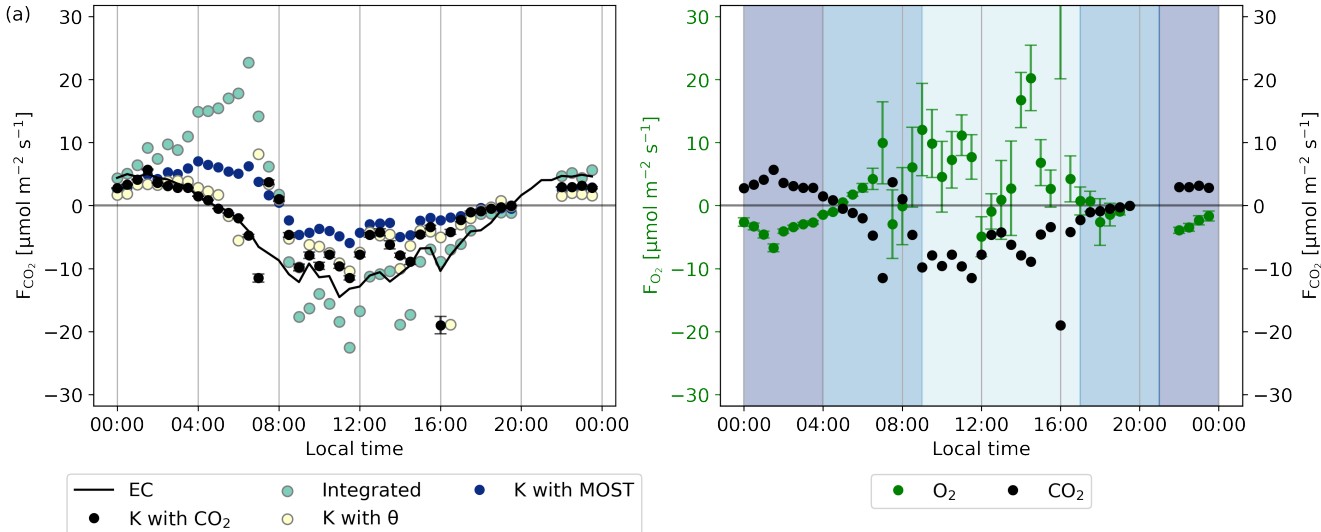

**Figure 6.** The $CO_2$ flux (a) calculated with different methods for the representative day, as described in Sect. 2.3, compared to the $CO_2$ flux of the ICOS EC measurements. (b) The comparison between the $O_2$ and $CO_2$ flux calculated using the method that gave the best results for the $CO_2$ flux calculations (using the exchange coefficient K with $CO_2$), for the representative day. The shaded colours indicate the regions that were selected for: the night signal (21:00-04:00), the day signal (09:00-17:00) and the remaining regions (04:00-09:00 and 17:00-21:00), with the time in local wintertime, +2UTC. The error bars of (b) are based on the error propagation of the standard error of the 30-minute values for the representative day, which are based on Eq. (4).

## 3.3 Flux calculations for $CO_2$ and $O_2$

We explored four alternative methods to derive the $O_2$ flux from the vertical gradient of the two measurement levels, as described in Sect. 2.3. Figure 6 shows both the theoretical and the observation-based approach that were used to calculate the $CO_2$ flux and the comparison with the ICOS EC $CO_2$ flux measurements at 27 m on the tower. By comparing these approaches to the EC measurements, we determined which method is most suitable to calculate the $O_2$ flux. The $CO_2$ flux measured by the EC system stays positive until around 05:00, when the respiration fluxes are the most dominant and the nocturnal boundary layer is shallower. After 05:00, the $CO_2$ flux of the EC system becomes negative, and the forest begins to take up $CO_2$ instead of emitting it. The assimilation fluxes increase and exceed the respiration fluxes, the boundary layer starts to grow and air with lower $CO_2$ concentrations is entrained from the free troposphere. After 20:00, the $CO_2$ flux of the EC system becomes positive again because the assimilation fluxes decrease, and the respiration signal begins to dominate again while the boundary layer height decreases. We expect to find this diurnal pattern and the sign change in our calculations of the $CO_2$ flux from the vertical gradient method as well.

First, we discuss the theoretical methods that are indicated in Fig. 6 with 'K with MOST' and 'Integrated' approach (see Sect. 2.3). The MOST and the Integrated method both overestimate the $CO_2$ flux during the night, between 0:00 and 05:00. In

addition the resulting $CO_2$ flux decreases and becomes negative too late in the day compared to the EC measurements. Both the $CO_2$ flux of the MOST and Integrated method evolve from a positive flux to a negative flux around 8:00. This is three hours later than the $CO_2$ flux from the EC measurements. During the day, between 08:00 and 15:00, the MOST method underestimates the $CO_2$ uptake and the Integrated method overestimates it. Table 2 shows that both MOST and the Integrated method have the highest mean difference and Root Mean Square Error (RMSE) compared to the observation-based approaches. We

discuss this further in Sect. 4.3. As result of this analysis, we decided to not use the theoretical approach to calculate the $O_2$ flux.

**Table 2.** The mean difference and the Root Mean Square Error (RMSE) of the comparison between the EC $CO_2$ flux measurements at 27 m in the tower and the $CO_2$ flux calculated with different methods for the exchange coefficient K, based on the ICOS data, each using the vertical gradient of $CO_2$ at 23 m and 125 m of our campaign data.

| Approach for K | Mean difference | RMSE |
|---|---|---|
| | [$\mu$mol m$^{-2}$ s$^{-1}$] | [$\mu$mol m$^{-2}$ s$^{-1}$] |
| Integrated | 5.21 | 7.81 |
| K with MOST | 4.98 | 5.83 |
| K with $\theta$ | 3.71 | 4.83 |
| K with $CO_2$ | 2.80 | 3.88 |

Secondly, we analyze the observation-based approaches, that are indicated in Fig. 6 with 'K with $\theta$' (where K is established using ICOS vertical gradients of potential temperature and the sensible heat flux) and 'K with $CO_2$' (where K is established using ICOS $CO_2$ vertical gradients and $CO_2$ EC data). The observation-based approaches showed a better comparison with the

EC observations in determining the $CO_2$ flux compared the to theoretical approach. Both the $\theta$ and the $CO_2$ method represent satisfactorily the nocturnal $CO_2$ flux between 0:00 and 5:00. After 5:00, the fluxes calculated by both methods start to decrease and change sign around the correct time (5:00) from a positive to a negative flux. During the day between 8:00 and 15:00, both the $\theta$ and the $CO_2$ methods underestimate the $CO_2$ flux, but not as much as the theoretical methods. Table 2 also shows that both the $\theta$ and the $CO_2$ methods have the lowest Mean difference and RMSE. Based on the smaller mean difference and

RMSE, and the direct link of $CO_2$ with $O_2$, we decided to proceed with the method where K is calculated with the ICOS data of $CO_2$, instead of the ICOS $\theta$ data. This K was then multiplied with our measured $O_2$ vertical gradient between 23 m and 125 m to calculate the $O_2$ flux. Section 4.3 presents a more complete discussion on the different methods to determine the most suitable K.

The resulting $O_2$ flux calculated with the exchange coefficient K based on the ICOS $CO_2$ data is shown in Fig. 6b. The uncertainties are based on the error propagation of the standard errors of the $O_2$ and $CO_2$ data per time step as calculated with Eq.

(4), in Eq. (7). We do not calculate an uncertainty for K, as this is not the dominating term contributing to the total uncertainty. The daytime flux values have a high variability, but the inferred fluxes appear physically realistic and promising for one of the first attempts to calculate $O_2$ fluxes. During the night, between 0:00 and 5:00, the $O_2$ flux data has a relatively stable negative

value, because the forest consumes $O_2$ for the respiration processes. Similarly $CO_2$ is released during the night, leading to a positive $CO_2$ flux. After 5:00, the $O_2$ flux becomes positive and shows a higher variability. Overall, the $O_2$ flux is positive during the day which indicates that the forest produces $O_2$ because the assimilation rate is higher then the respiration rate. The high variability of the $O_2$ flux compared to the $CO_2$ flux is caused by the less precise measurements of the $O_2$ vertical gradient compared to the $CO_2$ gradient (Fig. 4). The measurement precision needed to measure the difference between the two levels is

very high, and therefore impacts the measurement of the gradient of $O_2$. The nighttime values of the $O_2$ flux are therefore more reliable than the daytime values, since the difference between the two heights is larger and due to the more stable atmospheric conditions at night.

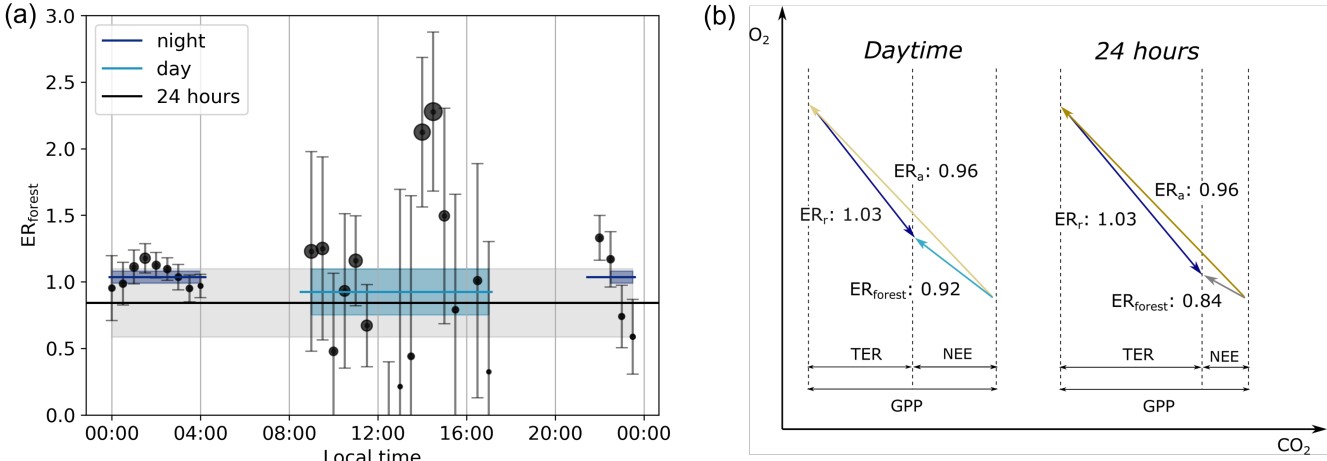

**Figure 7.** The half-hourly exchange ratio of the forest ($ER_{forest}$) and the resulting averaged $ER_{forest}$ for the entire day (black line), the night between 21:00-4:00 (dark blue line) and the day between 9:00-17:00 (light blue line), of the representative day (a) with the time in local wintertime, +2UTC. The size of the dots indicates the size of the absolute $O_2$ flux and the shaded bands indicate the uncertainties of the different $ER_{forest}$ signals. Note that the $ER_{forest}$ lines do not match with the average of the dots in the specific time period, because the lines are based on the averaged fluxes. These different ER signals are presented in a vector diagram format with the carbon fluxes, Gross Primary Production (GPP), Total Ecosystem Respiration (TER) and Net Ecosystem Exchange (NEE), and the ER of the assimilation processes ($ER_a$) and the ER of the respiration processes ($ER_a$) (b).

By using Eq. (6), we find three different $ER_{forest}$ signals throughout the day (Fig. 7 and Table 3). The selected time periods

based on the criteria described in Sect. 2.3 are between 09:00-17:00 for the daytime and between 21:00-04:00 for the nighttime (Fig. 6). This results in a nighttime $ER_{forest}$ signal of $1.04 \pm 0.04$, a daytime $ER_{forest}$ signal of $0.92 \pm 0.17$ and an $ER_{forest}$

signal for the entire 24 hours of $0.83 \pm 0.24$. Note that this 24h value is not the average of the day and night $ER_{forest}$ signals or from all the 30-minute $ER_{forest}$ signals, because we used the averaged fluxes. This means that the $ER_{forest}$ signals based on high flux values, indicated in Fig. 7 with larger symbols, contribute more to the averaged $ER_{forest}$ signals compared to the lower flux values. Figure 7b illustrates that when combining surface fluxes with different sign, we cannot just average the corresponding ER signals (see Sect. 4.4). The individual $ER_{forest}$ values of every 30-minutes show a clear difference between the day- and nighttime. The $ER_{forest}$ values during the nighttime are relatively stable. The $ER_{forest}$ values during the daytime show more variability, caused by the high variability of the $O_2$ flux during daytime (Fig. 6). The uncertainty of the $ER_{forest}$ signals is determined by the propagation of the standard error of the aggregate 30-minute data (based on Eq. (4)), in Eq. (7) and (6).

## 3.4 GPP and TER calculations

**Table 3.** The exchange ratio for the atmosphere ($ER_{atmos}$: Sect. 3.2), the forest ($ER_{forest}$: Sect. 3.3), and assimilation and respiration ($ER_a$ and $ER_r$: Sect. 3.3) for different time periods of the representative day. The time periods used to calculate the signals are: (09:00-13:00) for day and (23:00-04:00) for night of $ER_{atmos}$, and (09:00-17:00) for day and (21:00-04:00) for night of $ER_{forest}$, $ER_r$ and $ER_a$. Note that the uncertainty for $ER_{atmos}$ does not represent the same uncertainty as for $ER_{forest}$, since the first is the error of the fit, and the second is based on error propagation of the half hourly measurements.

|  | $ER_{forest}$ | $ER_r$ | $ER_a$ | $ER_{atmos}$ |
|---|---|---|---|---|
| Night | $1.03 \pm 0.05$ | $1.03 \pm 0.05$ |  | $1.22 \pm 0.02$ |
| Day | $0.92 \pm 0.17$ | $1.03 \pm 0.05$ | $0.96 \pm 0.12$ | $1.10 \pm 0.12$ |
| 24 hours | $0.84 \pm 0.26$ | $1.03 \pm 0.05$ | $0.96 \pm 0.11$ | $2.05 \pm 0.03$ |

We found the ER signals for assimilation ($ER_a$) and respiration ($ER_r$) by using Eq. (9) (Fig. 7b and Table 3). The assumption that $ER_r$ stays constant throughout the day seems reasonable, because the $ER_{forest}$ values stay stable during the night. A more elaborate discussion of this assumption can be found in Sect. 4.5. Therefore the $ER_r$ signal becomes $1.03 \pm 0.05$. $ER_a$ of the daytime is $0.96 \pm 0.11$, which indicates the $ER_a$ signal of the boreal forest when the surface fluxes are the highest. The $ER_a$ signal of the entire diurnal cycle is $0.95 \pm 0.11$, which also includes the assimilation processes during sunrise and sunset. Figure 7b shows all these ER signals and how they change throughout the day, together with their carbon fluxes. $ER_a$, $ER_r$ and the resulting $ER_{forest}$ signals are more realistic compared to the $ER_{atmos}$ signals. We will elaborate on these differences in Sect. 4.4 and 4.5.

By using Eq. (8) and (9) for a second representative day (13 through 15 June), with the $ER_a$ and $ER_r$ signals determined from the representative day, we show in Fig. 8 that the $O_2$ method compares well with the EC method. This means that the $O_2$ method could potentially be used to separate NEE into GPP and TER on any day where good simultaneous $CO_2$, $O_2$ and NEE measurements are available. The difference between the $CO_2$ fluxes determined with the $O_2$ method and the EC method

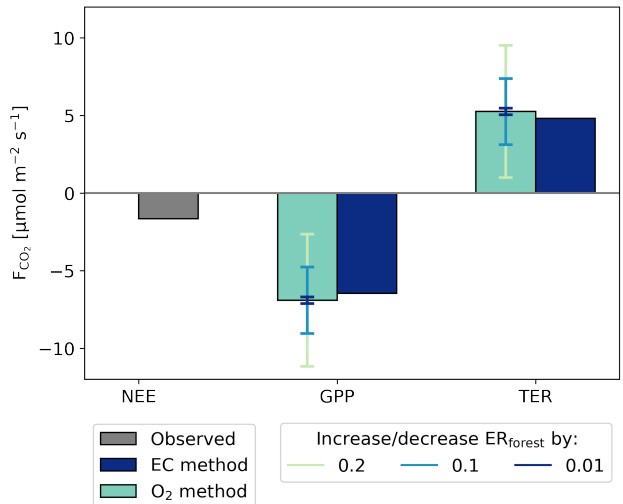

**Figure 8.** The $CO_2$ fluxes of a second representative day (13 through 15 June) for Net Ecosystem Exchange (NEE), Gross Primary Production (GPP) and Total Ecosystem Exchange (TER) based on two different methods: the EC method and the $O_2$ method. The different error bars indicate an increase/decrease of 0.2, 0.1 or 0.01 for the Exchange Ratio of the forest ($ER_{forest}$), used in the $O_2$ method.

of both the GPP and the TER flux are around 0.5 $\mu$mol m$^{-2}$ s$^{-1}$, which is less that 6% of the total gross flux. The difference is relatively small which means that the $O_2$ method compares well with the EC methods to separate NEE into GPP and TER. The different error bars in Fig. 8) show how sensitive the $O_2$ method is to the accuracy of $ER_{forest}$. By changing $ER_{forest}$ with 0.2, the GPP estimation by the $O_2$ method changes by 4 $\mu$mol m$^{-2}$ s$^{-1}$ and by changing $ER_{forest}$ with only 0.01, the GPP estimation changes with 0.2 $\mu$mol m$^{-2}$ s$^{-1}$. The effect of changing $ER_{forest}$ on TER has the same effect on GPP. This shows

that the $O_2$ method is quite sensitive to $ER_{forest}$ and should be measured accurately, with a suggested precision of around 0.05. With a precision of 0.05 for $ER_{forest}$ the GPP and TER fluxes derived with the $O_2$ method stay in the same range as the GPP and TER fluxes determined with the EC method. The application of the $O_2$ method will be further discussed in Sect. 4.5.

## 4   Discussion

We aimed to advance understanding of the $O_2 : CO_2$ exchange ratio and its diurnal variability over a boreal forest by continu-

ously measuring both $O_2$ and $CO_2$ concentrations at two heights above the canopy. These measurements gave us the possibility to compare the $ER_{atmos}$ and $ER_{forest}$ signal of an aggregate representative day and compare the boreal forest signals to previous studies in different ecosystems. Our $ER_{atmos}$ signal changed between the day (2.28) and the night (1.22) and had an overall diurnal signal of 2.05. For the $ER_{forest}$ signal, we needed to determine the $O_2$ and $CO_2$ surface fluxes based on the two heights. Different flux calculating methods were compared. The $O_2$ flux was calculated with the method that resulted in the best com-

parison to EC fluxes for $CO_2$, where we found that the exchange coefficient K based on the $CO_2$ data was most suited. The resulting $ER_{forest}$ signal showed again differences between the day (0.92) and night (1.04) and the overall diurnal $ER_{forest}$ was

0.83. For these differences and variability in the ER signals, different aspects of the uncertainty have to be taken into account, on which we elaborate in the next sections.

## 4.1 Measurement uncertainty

Analyzing the mean difference and standard deviation of the target cylinder values between 16-06-2019 and 17-07-2019 (Table 1), we see that the values are relatively high. Previous studies that used a fuel cell analyser for continuous atmospheric $O_2$ measurements (Battle et al., 2019; Ishidoya et al., 2013; van der Laan-Luijkx et al., 2010; Popa et al., 2010; Pickers et al., 2022), achieved measurement precision of around 5 per meg. WMO recommends a compatibility goal of 2 per meg, however this is difficult to achieve and therefore the extended compatibility goal is 10 per meg for the world-wide $O_2$ monitoring network (Crotwell et al., 2020), which shows that our long-term measurement precision of 19 per meg is relatively poor. This poor measurement precision could have been caused by several reasons; the $O_2$ values of the calibration cylinders that were used were relatively far apart, making it more difficult to measure the values around the target cylinder value. For 2018 we used calibration cylinders with the following values (on the SIO scale): -628.53 per meg -816.17 per meg, and -1208.28 per meg and for 2019 we used cylinders with values: -729.96 per meg, -816.17 per meg, and -1208.28 per meg. The cabin in which the instrument and cylinders were located was not well insulated, which created unstable temperature conditions which might have affected the stability of the cylinders (Keeling et al., 2007). The calibrations of our representative aggregate day took place during the night and therefore large temperature changes during the day might have affected daytime stability of the reference cylinder. Furthermore, tiny leakages in the setup might have influenced the measurements. Due to the relatively short period for these campaigns and remote location, it is not possible to trace back the cause of this large uncertainty. This high uncertainty resulted in a larger uncertainty of the vertical gradient of the two heights of the $O_2$ measurements. However, in this study we are mostly interested in the diurnal variability of the ER signal and differences between $ER_{atmos}$ and $ER_{forest}$ and therefore the long-term stability of the measurements are less relevant here compared to other $O_2$ studies.

To reduce the effect of the high measurement uncertainty and derive a more statistically robust signal of the vertical gradient, we created an aggregate representative day based on days with similar weather and atmospheric conditions. The increased statistics of this representative aggregate day decrease the effect of the low measurement precision. We also move away from the reality of one specific day, but rather focus on an average situation and variability of the ER signal above a boreal forest based on $O_2$ and $CO_2$ measurements at 2 levels. Given that very few previous studies focused on deriving forest ER signals globally, our analysis helps to gain further understanding of the diurnal variability and the difference between $ER_{atmos}$ and $ER_{forest}$, which will be discussed in the following sections.

## 4.2 $ER_{atmos}$ signal in comparison to previous studies

Despite the uncertainty in our measurements, there are clear differences between the slopes of $O_2$ and $CO_2$ throughout the diurnal cycle (Fig. 5). Three different $ER_{atmos}$ signals are visible, with two signals for the day ($2.28 \pm 0.01$ and $1.10 \pm 0.12$) and one for the night night ($1.22 \pm 0.02$) slope (Table 3). Note that the uncertainty of these values is based on the slope of

the fitted line in Fig. 5 and does not represent the uncertainty in the stability of our measurements indicated in Table 1. The difference between day and night values of $ER_{atmos}$ was expected because different processes (i.e. respiration, assimilation and entrainment) with different ER signals play a role at different times during the diurnal cycle. To exclude as much as possible the effect of entrainment and the boundary layer dynamics during the morning transition, we will from now on refer to the 1.10 value as the day $ER_{atmos}$ signal, which is the signal derived form period P3a. $ER_{atmos}$ for the complete day results in $2.05 \pm 0.03$.


**Table 4.** The different Exchange Ratio (ER) signals of previous studies, with the ER of the atmosphere ($ER_{atmos}$), the ER of the forest ($ER_{forest}$), the ER of the respiration processes ($ER_r$) and the ER of the assimilation processes ($ER_a$). The different studies are: Bat, 2019: (Battle et al., 2019), Ish, 2015: (Ishidoya et al., 2015), Ish, 2013: (Ishidoya et al., 2013), Sei, 2004: (Seibt et al., 2004).

| Study | $ER_{atmos}$[a] | | | $ER_{forest}$[b] | | | $ER_r$ | $ER_a$ |
| | Day | Night | 24 hours | Day | Night | 24 hours | | |
|---|---|---|---|---|---|---|---|---|
| This study | $1.10 \pm 0.12$ | $1.22 \pm 0.02$ | $2.05 \pm 0.03$ | $0.92 \pm 0.17$ | $1.03 \pm 0.05$ | $0.84 \pm 0.26$ | $1.03 \pm 0.05$ | $0.96 \pm 0.12$ |
| Bat, 2019 | $1.02 \pm 0.01$ | $1.12 \pm 0.01$ | | | | | | |
| Ish, 2015 | | | | $< 1.0$ | $> 1.0$ | $0.86 \pm 0.04$ | $1.11 \pm 0.01$ | $1.0$ |
| Ish, 2013 | $0.87 \pm 0.02$ | $1.03 \pm 0.02$ | $0.94 \pm 0.01$ | $\approx 0.98$ | $\approx 1.11$ | $0.89$ | $1.11 \pm 0.01$ | $1.02 \pm 0.03$ |
| Sei, 2004[c] | | | $1.01 \pm 0.06$ | $1.24 \pm 0.06$ | $1.01 \pm 0.02$ | $1.26 \pm 0.05$ | $0.94 \pm 0.04$ | $1.19 \pm 0.12$ |
| Sei, 2004[d] | $1.14 \pm 0.19$ | $1.16 \pm 0.02$ | $1.03 \pm 0.05$ | | | | | |

[a] An ER signal is classified as $ER_{atmos}$ when the ER signal is based on one concentration measurement of $O_2$ and $CO_2$.

[b] An ER signal is classified as $ER_{forest}$ when the ER signal is based on surface fluxes from either an 1-box model or vertical gradient flux calculations.

[c] The ER signals of the location Griffin Forest of Seibt et al. (2004) are used here.

[d] The ER signals of the location Harvard Forest of Seibt et al. (2004) are used here.

When comparing our $ER_{atmos}$ signals to those from Battle et al. (2019), Ishidoya et al. (2013) and Seibt et al. (2004) (Table 4), we note several similarities but also some differences regarding the specific values of the $ER_{atmos}$ signals. Our daytime signal of 1.10 is similar to 1.02, 0.87 and 1.14 from the previous studies respectively, as is our nighttime signal of 1.22 compared to 1.12 (Battle et al., 2019), 1.03 (Ishidoya et al., 2013) and 1.16 (Seibt et al., 2004). However, our 24-hours $ER_{atmos}$ signal of 2.05

shows an unrealistically high number which clearly does not indicate the ER of the forest only. A typical $ER_{atmos}$ signal for a 24 hour period lies around 1, as is shown in table 4 and by Stephens et al. (2007) and Manning (2001). Our 24-hours $ER_{atmos}$ value includes the measurement points of the period that is influence by entrainment and boundary layer dynamics (P2), for which period we found an ER signal of 2.28. The large influence of entrainment and boundary layer dynamics made it difficult to be very precise about the specific time periods to choose for P3. Moving the selected time boundaries of P3a from 9:00

to 9:30 or from 13:00 to 12:30 leads to $ER_{atmos}$ values of 0.88 or 1.75 respectively. The large changes in the daytime $ER_{atmos}$ due to small changes in the time boundaries, shows the high uncertainty of the daytime $ER_{atmos}$. Therefore, our measurements provide a confirmation of earlier indications (Seibt et al., 2004) that $ER_{atmos}$ is an unreliable estimate for the ER of a forest, and we recommend to use $ER_{forest}$.

Instead, $ER_{atmos}$ also represents how $O_2$ and $CO_2$ are influenced by the boundary layer dynamics and entrainment (Fig. 1). The high $ER_{atmos}$ values cannot be explained by signals from other sources, such as fossil fuel combustion or exchange with the lake, as both are not represented in the footprint of our measurements (see Sect. 2.3). Furthermore, we have shown that these high values are not an artefact from the instability of the pressure stabilization, as preliminary analysis of the $ER_{atmos}$ values from our 2018 measurements also show values higher than 2.0 (not shown). Although we cannot fully rule out remain-

ing artefacts in the calibration due to e.g. temperature changes in the measurement cabin, we suggest that the more plausible explanation is that $ER_{atmos}$ is highly influenced by atmospheric processes, such as entrainment. The entrainment of air from either the residual layer (early in the morning transition) or the free troposphere (after the residual layer is dissolved) could impact the $ER_{atmos}$ as different sources of air are mixed. The residual layer contains air from the day before and could be affected by horizontal advection whereas the air in the free troposphere originates from different background sources. These difference

sources can have different ER signals and therefore create a mixture of air where $O_2$ and $CO_2$ are influenced differently. These air masses affect $O_2$ differently compared to $CO_2$ in the boundary layer and an $ER_{atmos}$ signal will arise that cannot be linked directly to one specific process. Even though entrainment processes also occur at locations of previous studies, we still find differences in $ER_{atmos}$. We suggest that this can be explained by difference in measurement height compared to the canopy height and different sources of background air in the free troposphere at the measurement location. For the $ER_{atmos}$ signal

during P2 at 125 m, we find a value of 3.40, even higher than the $ER_{atmos}$ signal of 2.28 at 23 m, which indicates the influence of entrainment increases when measuring further away from the canopy and as a result the $ER_{atmos}$ signals shows higher values. Further insights on the contributions of each process to $ER_{atmos}$ cannot be estimated from the measurements alone, and would require using an atmospheric model.

### 4.3    Uncertainties in the $CO_2$ and $O_2$ flux calculations

By comparing the theoretical and observation-based methods, we determined that the most suitable method to calculate both the $CO_2$ and $O_2$ flux was to use the observation-based method with $CO_2$ data (Sect. 3.3). Figure 6 and Table 2 show that the theoretical methods (MOST and Integrated) resulted in a change of the $CO_2$ flux that was late compared to the EC-measurement. This delay has been described before and is caused by the time it takes before the turbulence can mix the $CO_2$ gradient

driven by stable nocturnal stratification conditions and establish the corresponding gradient to how turbulent the atmosphere is (Casso-Torralba et al., 2008). When the heights of the gradient are closer together, the delay is less pronounced. However, the measurement heights used during our campaign are relatively far apart (125 m and 23 m) and the EC flux is measured at 27 m. The 125 m measurement is even located outside the surface layer during the morning transition. This made the flux-gradient method (as described in Eq. (7)) less applicable, which assumes that the surface flux stays constant in the surface layer (Dyer,

580    1974).

Since during our campaign we only measured at two heights, we missed information on the logarithmic profile originating from the canopy top, which resulted in an underestimation of the flux using the K with MOST. This was solved by integrating

the MOST equation ('Integrated method'). With the integrated method, the gradient is assumed to be logarithmic and the total flux increases compared to the MOST calculation (Paulson, 1970). However, with the large difference between the two measurement heights, the integrated approach still overestimated the $CO_2$ flux compared to the EC measurements during both the day and the night. Also, the delay in the timing of the sign change of the gradient cannot be solved with this Integrated method. We also explored the effect of adding a Roughness Surface Layer (RSL) in the flux calculations of the theoretical methods, by adding an extra factor that accounts for this layer (not shown in the results) (de Ridder, 2010). The contribution of the RSL did not improve our results, because it also includes the delay of the gradient which was causing the largest deviation in the theoretical methods (Table 2).

By applying both observation-based methods, using either $\theta$ or $CO_2$ to infer the exchange coefficient K, we did not find this delay in the timing of the gradient and the observation-based methods therefore resulted in derived fluxes close to the EC measurements. Here it has to be noted that the ICOS EC measurements of $CO_2$ that we used as a benchmark for the most suitable flux calculation approach, was also used in calculating K with $CO_2$, which makes the comparison of these approaches to the $CO_2$ flux not fully independent. Note that we first derive K with the vertical $CO_2$ gradients calculated from ICOS $CO_2$ observations at three vertical levels, and apply this to our own measurements of the $CO_2$ vertical gradient with an independent instrument (Table B1). As a result there is not a full circularity when comparing the obtained fluxes to the EC $CO_2$ measurements to select which method for calculating K we use. Most previous studies that determined fluxes based on the gradient-approach used $\theta$ to calculate K (Stull, 1988; Mayer et al., 2011; Wolf et al., 2008; Bolinius et al., 2016; Brown et al., 2020), because $\theta$ is the driver of convective turbulence. However, because $O_2$ is directly linked to $CO_2$ and our statistics (Table 2) indicated that the $CO_2$ method resulted in a better comparison to the EC fluxes, we decide to use the ICOS $CO_2$ data at 3 levels and the $CO_2$ EC measurements to calculate K. This K together with the measurements of two heights by our instrument during our campaign were used to calculate both the $CO_2$ and the $O_2$ fluxes used in our study. We also tested the impact of using only 2 vertical levels of the ICOS $CO_2$ concentrations to calculate K (not shown), which was also the case in the only previous study that derived $O_2$ fluxes. Ishidoya et al. (2015) derived $O_2$ fluxes for a temperate forest in Japan using 2 vertical levels at 18 and 27 m height for both $O_2$ and $CO_2$ concentrations. Our comparison of deriving K based on 2 vertical levels (23 m and 125 m), resulted in an underestimation of the gradient and thus an overestimation of K, and as a consequence the calculated $CO_2$ flux was overestimated. Therefore the 3 levels of ICOS $CO_2$ concentrations measurements proved to be vital in our flux calculations here. We still missed the logarithmic profile at the surface with only the two vertical campaign measurements and as a result slightly underestimated the final $CO_2$ and $O_2$ flux. Therefore, we recommend to always measure at at least three heights of $CO_2$ and $O_2$ inside the surface layer, when they are meant to be used for flux calculations.

Our final $O_2$ flux (Fig. 6) shows a clear diurnal cycle, with the expected behaviour of negative values in the night ($O_2$ consumption for respiration) and a positive flux during the day ($O_2$ release during assimilation). The nighttime fluxes are more stable and give a clear signal due to the larger vertical gradient. K is more difficult to determine during the night because the EC measurements are less representative due to the low level of turbulence. However, the largest contributor to the uncer-

tainty are our own $O_2$ measurements and the larger gradient allows to better establish the $O_2$ flux. The larger variability of the

daytime $O_2$ fluxes is caused by the smaller gradient of the $O_2$ concentration measurements during the day (Fig. 3), when the atmosphere is more well-mixed and the difference between the two heights becomes smaller. The relatively large measurement uncertainty made it difficult to measure these small difference between the two heights and increased the noise in the fluxes. The measurement noise resulted in $O_2$ gradient variations that were not tied to the $CO_2$ gradient variations and this degraded the correlation between the two fluxes. Despite this larger variability, we still find a clear diurnal behaviour, which allowed

us to calculate $ER_{forest}$. Note that the uncertainties of the surface fluxes of $O_2$ and $CO_2$ are only based on the measurements from our campaigns and we did not include the uncertainties that are related to the calculations of K based on the ICOS data. However, the uncertainty in K is relatively small compared to the other terms in the calculation, and the final uncertainty of estimates is dominated by the measurement uncertainty of $O_2$. Omitting the uncertainty associated with K therefore leads to a minor underestimate of the full uncertainty.


### 4.4 $ER_{forest}$ signal compared to previous studies

Our resulting $ER_{forest}$ signal changes throughout the diurnal cycle, with specific daytime ($0.92 \pm 0.17$), nighttime ($1.03 \pm 0.05$) and overall ($0.84 \pm 0.26$) values (Fig. 7 and Table 3). The individual nighttime values show a smaller uncertainty due to the already explained effect of the larger gradient during the stable atmospheric conditions of the night. In contrast, the individual

daytime values show a larger uncertainty due to the smaller gradient during the unstable atmospheric conditions of the day. We therefore used averaged values for the daytime and nighttime signals to derive the $ER_{forest}$ values. While the daytime signal excludes the entrainment and the boundary layer dynamics during the morning transition, these effects are still included in the overall $ER_{forest}$ signal. Note that the overall 24 hour signal is not the average of the daytime and nighttime signal. The nighttime $ER_{forest}$ signal represents a negative $O_2$ flux and a positive $CO_2$ flux, whereas the daytime $ER_{forest}$ signal represents a positive

$O_2$ flux and a negative $CO_2$ flux. This means that the daytime and nighttime surface fluxes influence the atmosphere differently and therefore these $ER_{forest}$ values cannot be averaged to calculate the overall $ER_{forest}$ signal. By first calculating the average overall $O_2$ and $CO_2$ fluxes and then divide these, we derive the overall $ER_{forest}$ signal correctly.

When comparing our $ER_{forest}$ signals to previous studies of Seibt et al. (2004), Ishidoya et al. (2013, 2015) (Table 4) we

notice that the difference between the daytime and the nighttime values that we found and the specific values of the different $ER_{forest}$, have some similarities and some differences. Our results, along with these of Ishidoya et al. (2013, 2015) (night: 1.11 and day: 0.98) show the $ER_{forest}$ signal of the nighttime is being higher than the daytime signal, whereas Seibt et al. (2004) (day: 1.24 and night: 1.01) showed the opposite behaviour. Our results are most similar to the signals of both Ishidoya et al. (2013) and Ishidoya et al. (2015), especially if we take our uncertainty range into account. The 24 hours signals are difficult to

compare as we used a different method to determine the overall $ER_{forest}$ signal compared to Ishidoya et al. (2015). In this study we use average fluxes instead of a linear regression through either the $O_2$ and $CO_2$ fluxes or vertical gradient and we thereby take into account the size of the fluxes that contributes most to the ER signal, which results in a flux-weighted average $ER_{forest}$.

We note again that we need to distinguish between daytime and nighttime signal, and we cannot just average them. Figure 7 illustrates the need to take averages in consistent meteorological and biological periods that are characterized by similar

turbulence regimes and similar signs of the $O_2$ and $CO_2$ exchange. For example, combining a small negative $O_2$ flux with a high ER, with a large positive $O_2$ flux with a lower ER, results in a smaller $O_2$ flux, compared to when the ERs of both fluxes would have been the same. When we take into account our uncertainty, the complete day signal of $0.84 \pm 0.26$ comes close to the globally used average ER of the biosphere of 1.1 (Severinghaus, 1995). However, the specific value suggest that the overall $ER_{forest}$ signal of this boreal forest lies somewhat lower than 1.1, closer to 1.0. The difference in $ER_{forest}$ signals between studies

can be explained with the different $ER_a$ and $ER_r$ signals, which we discuss in Sect. 4.5.

The $ER_{forest}$ and $ER_{atmos}$ signals are not identical, and they do therefore not represent the same information (Table 3). The $ER_{atmos}$ signals are higher compared to the $ER_{forest}$ signals, especially the 24-hour signals show a large difference. Despite the higher numbers, the day and night signals of $ER_{atmos}$ and $ER_{forest}$ show both the same pattern, where the daytime signal

is lower compared to the nighttime signal. When comparing these differences to previous studies we find that not all studies find the same results. The difference between $ER_{forest}$ and $ER_{atmos}$ was not found by Ishidoya et al. (2013). In contrast, Seibt et al. (2004) found a difference between $ER_{forest}$ and $ER_{atmos}$ (Table 4). A reason for this could be the measurement height of $ER_{atmos}$. When $ER_{atmos}$ is determined closer to the canopy and inside the roughness sublayer, it will be more influenced by the surface processes compared to measurements at higher levels, which are seeing more integrated signals of all processes that

influence the concentrations inside the atmospheric boundary layer (i.e. forest exchange and non-local processes like entrainment). To get a clear answer to this question, we should further investigate to what extent $ER_{atmos}$ is influenced by entrainment and boundary layer dynamics and under which conditions they can come close to $ER_{forest}$. We already show that excluding the morning transition (P2) helps to improve the $ER_{atmos}$ signal. However, as already stated, it is difficult from the measurements alone to determine if the $ER_{atmos}$ signal is influenced only by the surface during this period. An Atmospheric model would

therefore be needed to find how $ER_{atmos}$ can be derived from a single measurement height, and allow comparison to previous studies that measured at one height to determine the ER of the forest (Battle et al., 2019; van der Laan et al., 2014; Stephens et al., 2007). We are currently applying a specific mixed-layer atmospheric model to further investigate this.

## 4.5 The $ER_a$ and $ER_r$ signals

To further understand the relationship between $O_2$ and $CO_2$, we cannot use the $ER_{forest}$ signal alone. To look in more detail into the processes driving the variations, we calculated the exchange ratios of respiration ($ER_r$) and assimilation ($ER_a$) (Table 3 and Fig. 7). $ER_r$ was taken as the $ER_{forest}$ night-time signal ($1.03 \pm 0.05$), by assuming that only respiration influences the $ER_{forest}$ signal during the night and that the $ER_r$ signal stays constant throughout the entire day. This means that both the heterotrophic and autotrophic respiration are included in $ER_r$ and the same components are respired in the same ratios throughout the day

to keep $ER_r$ a constant value. The studies that looked at the $ER_r$ of an ecosystem (Hilman et al., 2022; Angert et al., 2015; Hicks Pries et al., 2020) only focused on longer time scales than the diurnal cycle. It is therefore not possible to derive diurnal

variability of $ER_r$ from these previous studies. We would expect some changes in $ER_r$ as temperatures changes during the day and the respiration of plants involves photorespiration during daytime and dark respiration during nighttime. However, as the study of Hilman et al. (2022) showed, the variability of $ER_r$ mainly depends on the bulk soil respiration and therefore depends on the soil temperature and soil moisture. No large changes in temperatures or soil moisture were detected during the period of the representative aggregate day, and therefore it is unlikely that the $ER_r$ significantly changed in that period. To get a more detailed view on how the $ER_r$ of an ecosystem changes throughout the day, more research is needed on the variability of $ER_r$ including from plant respiration by chamber measurements.

The variability of $ER_r$ between locations highly depends on the soil properties (Angert et al., 2015), which makes it difficult to compare with the few studies available (Seibt et al., 2004; Ishidoya et al., 2013) that measured $ER_r$ with chamber measurements on a brown soil. The soil in our study area is a podzol, which is characterised by a high acidity with little organic matter (Buurman and Jongmans, 2005). The OR of podzols is around 1.08 (Worrall et al., 2013) and the ER of acid soils is expected to be around this OR, because carbon cannot easily dissolve into the groundwater (Angert et al., 2015), and we therefore conclude that our $ER_r$ value of 1.04 is realistic.

We looked at two options to calculate $ER_a$; $ER_a$ based only on the daytime measurements (between 9:00 and 17:00: 0.96 $\pm$ 0.12) and $ER_a$ based on all the measurement throughout the 24-hour period: 0.96 $\pm$ 0.11). Both numbers are close to 1, which is often assumed as a standard value for $ER_a$ (Ishidoya et al., 2015; Severinghaus, 1995). Next to that, a value of $ER_a$ close to 1 means that ammonium is used as a source for nitrogen, instead of nitrate (Bloom et al., 1989, 2012). Ammonium is indeed a larger source for nitrogen compared to nitrate in Hyytiälä (Korhonen et al., 2013). The OR of needle leaves, and plant material in general, appears to be always close to 1.0 (Jürgensen et al., 2021), which again confirms our $ER_a$ signals. We did not observe differences between the two $ER_a$ signals. The transition periods between the night and the daytime were difficult to measure, because the gradient then becomes close to zero, which means there could be a possibility that next to $ER_{forest}$, $ER_a$ also has a diurnal cycle. Again, there are only a few studies that looked at the variability of $ER_a$ (Fischer et al., 2015; Bloom et al., 1989, 2012; Bloom, 2015). The available studies show that $ER_a$ depends on light (Fischer et al., 2015) and the source of nitrogen in the soil (Bloom, 2015). These changes in $ER_a$ happen when the changes in the atmosphere and the soil are sudden and persist for a longer time compared to a diurnal cycle. We can therefore say that the $ER_a$ also does not change drastically during the day. To get a more detailed overview of $ER_a$, more precise measurements are needed with uncertainties lower than 0.1 for the ER signals. However, the similar values of $ER_a$ that we find for the daytime and 24 hour measurements show that $ER_a$ is hardly affected by entrainment during the morning transition and it would suggest that the morning transition is less of an issue for $ER_{forest}$ than for $ER_{atmos}$.

By applying the $O_2$ method to a new aggregate day, we showed that the $O_2$ method gives results similar to the EC method to partition NEE and derive the GPP and TER fluxes (Fig. 8), with estimates of the uncertainties of the $O_2$ method. The EC method also contains uncertainties in its approach because of the reliance on a function of temperature, and should therefore

not necessarily be assumed to be 'truth' (Reichstein et al., 2005). Despite the uncertainty of both the $O_2$ method and the EC method, both methods give similar results for the $CO_2$ flux of GPP and TER. In our comparison of the $O_2$ and EC methods, there is a minor degree of circularity, as we use the EC GPP estimates to estimate $ER_a$. By applying it to another representative day, we prevent a full circularity (Table B1). In this campaign, we unfortunately could not determine the $ER_a$ and $ER_r$ signals independently from EC, which would be recommended for a full comparison. This would have been possible by using chamber measurements. We expect only minor changes in $ER_a$ from branch/leaf chamber measurements compared to the values we derived, because our $ER_a$ is in the range of expected values compared to previous studies (e.g. Jürgensen et al. (2021); Bloom (2015); Fischer et al. (2015)). The satisfactory comparison between the $O_2$ and the EC methods for the partitioning of the fluxes shows the potential of the $O_2$ method. The largest challenge for this method is to determine $ER_{forest}$ with large enough accuracy, as this value is most variable and most difficult to determine based on the small $O_2$ gradients that we observed. Figure 8 shows that the $ER_{forest}$ signal should be measured with an uncertainty of 0.05 or less to get results within the uncertainty range of the EC method. When such high accuracy is reached, the $O_2$ method has the potential to provide an alternative method for the separation of GPP and TER without relying on the regularly used temperature-based function as used for the EC method. Ishidoya et al. (2015) showed similar results, where the $O_2$ method also produced GPP and TER comparable to the EC method and the magnitude of the GPP and TER fluxes highly depended on the derived $ER_a$ and $ER_r$ signals.

To allow an independent comparison between the flux partitioning with the EC method and the $O_2$ method, such as was done with using $\delta^{13}CO_2$ by Wehr et al. (2016), we recommend measuring the $ER_r$ and $ER_a$ signals directly with chamber measurements (Seibt et al., 2004; Ishidoya et al., 2013). We also recommend to add at least one additional measurement height for the $O_2$ and $CO_2$ concentrations below the canopy to apply the storage correction for both the $O_2$ and the $CO_2$ fluxes (Aubinet et al., 2012) and to add a measurement in the free troposphere to better evaluate the effect of entrainment. Despite the high dependency on the accuracy of the ER, this study showed again, as did (Ishidoya et al., 2015), that the $O_2$ method can be used to get a better understanding of the carbon cycle. To further develop this method we need to expand the $O_2$ measurements for longer time series and more locations, and analyze how $ER_{forest}$ varies over longer time scales, which can improve the global average value of ER ($\alpha_B$) of 1.1 as used in global carbon budget studies such as Manning and Keeling (2006).

## 5 Conclusions

By continuously measuring atmospheric $O_2$ and $CO_2$ concentrations at two heights in Hyytiälä, Finland, we gained new insights into the diurnal variability of $O_2$ and $CO_2$ above a boreal forest, quantified by interpreting their Exchange Ratio (ER). We showed that the signal based on one measurement height of the $O_2$ and $CO_2$ concentrations ($ER_{atmos}$) is not representative for the exchange between the forest and the atmosphere only, but instead includes other processes such as entrainment as well. To derive the ER of the forest specifically ($ER_{forest}$), we first determined the surface fluxes above the canopy of $O_2$ and $CO_2$ using the vertical gradient between the two measurement heights. We found that the most suitable method to calculate both

the $O_2$ and $CO_2$ surface fluxes was to use the exchange coefficient calculated from the Eddy Covariance (EC) $CO_2$ flux and the vertical gradient of $CO_2$ measurements at three heights above the canopy. The $ER_{forest}$ signals that resulted from the ratio of the mean $O_2$ and $CO_2$ fluxes varied between the daytime ($0.92 \pm 0.17$ mol/mol) and nighttime ($1.03 \pm 0.05$ mol/mol). The different $ER_{forest}$ signals were composed of the ER of respiration ($ER_r$: $1.03 \pm 0.05$ mol/mol) and the ER of assimilation ($ER_a$: $0.96 \pm 0.12$ mol/mol). With these findings we show improved methods to derive $O_2$ forest fluxes and to derive the variability in the different ER signals over a representative diurnal cycle. The $ER_{forest}$ signal shows a clear diurnal cycle for this boreal forest and the overall ratio is lower than 1.1 that is used in global carbon budget calculations. Finally, we show that these ER signals can be used to separate Net Ecosystem Exchange (NEE) into Gross Primary Production (GPP) and Total Ecosystem Respiration (TER).

With only a few data sets of continuous measurements of both $O_2$ and $CO_2$ concentrations over forests, our data set is of high importance, specifically the availability of measurements at two heights that allow calculation of $O_2$ and $CO_2$ fluxes. Our analyses can serve as a starting point for follow up research using coupled land surface-atmosphere models to distinguish and quantify contributions of different processes to $ER_{atmos}$ and $ER_{forest}$ signals. Further understanding of these differences will help to fully exploit the advantages of atmospheric $O_2$ when unraveling the different components in the carbon cycle.

*Data availability.* The data used in this study are available from https://doi.org/10.18160/SJ3J-PD38.

## Appendix A: Equations to calculate the Exchange Coefficient, K

### A1 Observation-based method

The gradient between three points is calculated with the following equation:

$$\overline{\phi}(z) = a \cdot z^2 + b \cdot z + c \tag{A1}$$

$$\left( \frac{\partial \overline{\phi}(z)}{\partial z} \right) = 2 \cdot a \cdot z + b \tag{A2}$$

Where z [m] is the height above the displacement height (d [m]) (d is taken as: $2/3 \cdot$ canopy height), $\overline{\phi}$ is the average variable where the line is fitted through and a, b and c are the resulted fitted parameters. When only two vertical measurements are available, the gradient was determined using finite differences.

### A2 Theoretical approach

For the MOST method, the following equations were used (Physick and Garratt, 1995):

$$K = \frac{\kappa \cdot z \cdot u_*}{\Phi_H(\frac{z}{L}) \phi_{rsl}(\frac{z}{L})} \tag{A3}$$

Where K is the exchange coefficient [m$^2$ s$^{-1}$], $\kappa = 0.4$ is the von Kármán constant, $u_*$ [m s$^{-1}$] is the friction velocity, $\Phi_H$ [-] indicates the stability function and $\phi_{rsl}$ [-] indicates the contribution of the roughness sublayer (RSL). The $\Phi_H$ was calculated with (Dyer, 1974):

$$\Phi_H\left(\frac{z}{L}\right) = \left(1 - 16\frac{z}{L}\right)^{-1/2} \qquad \text{when z/L} < 0 \tag{A4}$$

$$\Phi_H\left(\frac{z}{L}\right) = 1 + 5\frac{z}{L} \qquad \text{when z/L} > 0 \tag{A5}$$

Where L [m] is the Obukov Length, which was based on the following equation (Dyer, 1974):

$$L = \frac{-u_*^3}{\kappa(\frac{g}{\theta_v})(\overline{w'\theta_v'})} \tag{A6}$$

Where $\theta_v$ [K] is the virtual potential temperature, $\overline{w'\theta_v'}$ [K m s$^{-1}$] is the virtual surface heat flux and g [m s$^{-2}$] is the acceleration due to gravity. Because the flux was measured close to the canopy, the roughness surface layer (RSL) could become important. The RSL needs an additional length scale ($\phi$) and can be calculated with the following equation (de Ridder, 2010):

$$\phi_{HRSL}\left(\frac{z}{z_*}\right) = 1 - e^{-\mu\frac{z}{z_*}} \tag{A7}$$

Here $z_*$ [m] indicates the height of the RSL above the displacement height and we take that as (2 · canopy height-d) and $\mu$ is a constant of 0.95 [-].

By integrating Eq. (7) with Eq. (A3) for K, we get the following equation that was used for the Integrated method (Physick and Garratt, 1995):

$$\phi(z_2) - \phi(z_1) = \frac{(\overline{w'\phi'})}{\kappa \cdot u_*}\left[ln\left(\frac{z_2}{z_1}\right) - \Psi_H\left(\frac{z_2}{L}\right) + \Psi_H\left(\frac{z_1}{L}\right) + \psi_{RSL}\left(\frac{z}{L}, \frac{z}{z_*}\right)\right] \tag{A8}$$

Where $\Psi_H$ [-] are the integrated stability functions for heat and $\psi$ [-] is the integrated function to account for the roughness sublayer (RSL) effect. $\Psi_H$ was calculated with (Paulson, 1970):

$$\left.\begin{array}{l}\Psi_H\left(\frac{z}{L}\right) = 2ln\left(\frac{1+x^2}{2}\right) \\ \qquad x = (1 - 16z/L)^{1/4}\end{array}\right\} \qquad \text{when z/L} < 0 \tag{A9}$$

$$\Psi_H\left(\frac{z}{L}\right) = -5\frac{z}{L} \qquad \text{when z/L} > 0 \tag{A10}$$

The function of the integrated RSL length scale ($\psi_{RSL}$) [-] was calculated with (de Ridder, 2010):

$$\psi_{RSL}\left(\frac{z}{L}, \frac{z}{z_*}\right) \approx \Phi_H\left[\left(1 + \frac{\nu}{\mu z/z_*}\right)\frac{z}{L}\right]\frac{1}{\lambda}ln\left(1 + \frac{\lambda}{\mu z/z_*}\right)e^{-\mu z/z_*} \tag{A11}$$

Where $\nu$ and $\lambda$ are both parameters, taken as 0.5 and 1.5 respectively.

## Appendix B: Figures and tables

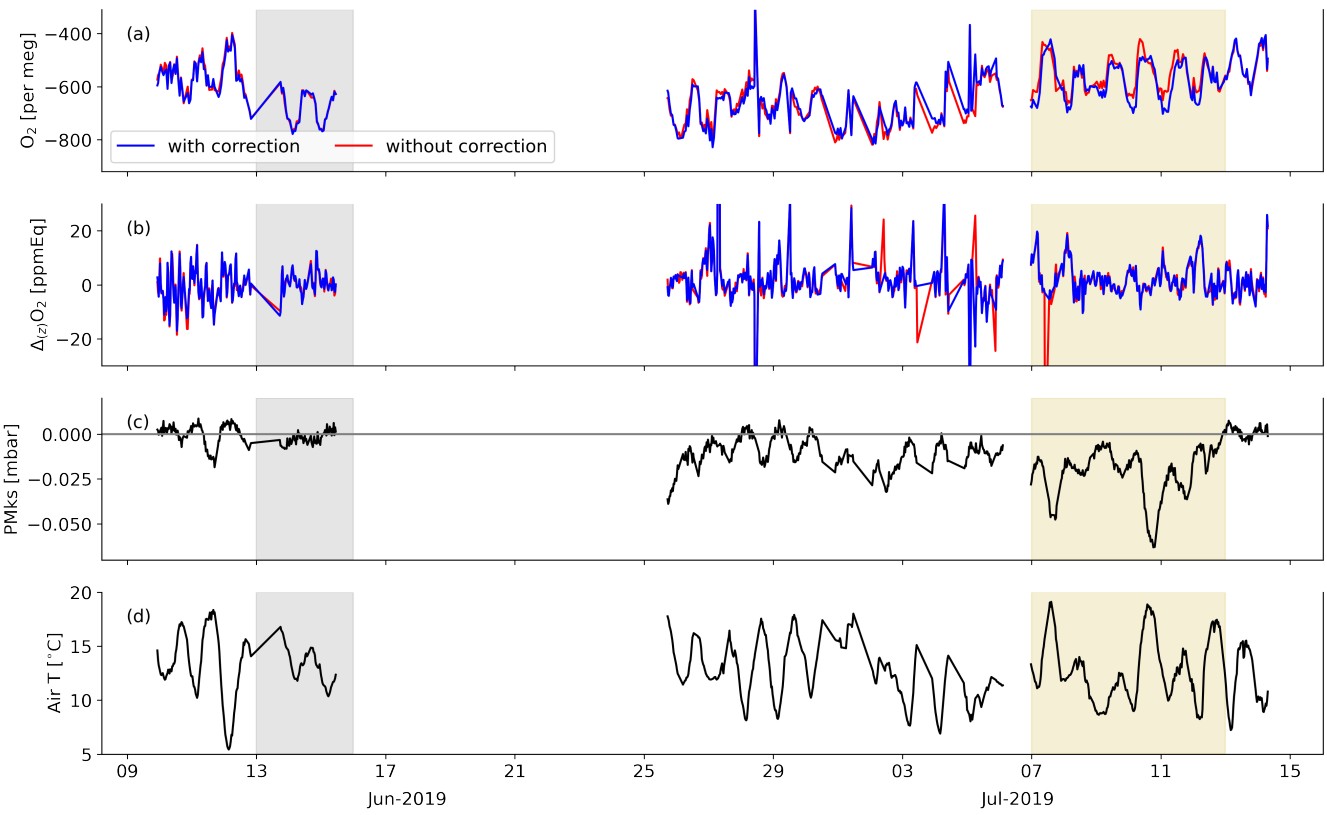

**Figure B1.** The corrected and uncorrected half-hourly average $O_2$ concentrations at 23 m for the PMKS deviations (a), together with the corrected and uncorrected gradient of $O_2$ (b). The time series for the 2019 measurement period of the PMKS (c) and the air temperature (d) are also shown. The shaded areas indicates the dates that were selected for the two aggregate representative days, 13 through 15 June (grey) and 7 through 12 July 2019 (yellow).

**Table B1.** The data used to calculate different variables that we calculated in our study for the two aggregate days. (a) indicates the data that was used for the first representative aggregate day between 7 through 13 July 2019 and (b) indicates the data that was used for the second aggregate day between 13 through 15 June 2019. The data used are mostly from the period of the respective aggregate day, except when indicated otherwise.

| (a) | Data used during the analysis of the aggregate day between 7 through 12 July 2019 [a] | | | | | |
|---|---|---|---|---|---|---|
| | (1) $ER_{atmos}$ | (2) Exchange coefficient (K) | (3) Surface flux of $O_2$ and $CO_2$ | (4) $ER_{forest}$ | (5) $ER_a$ and $ER_r$ | (6) GPP and TER |
| Data used: | · $O_2$ from campaign at 23 m  · $CO_2$ from campaign at 23 m | · EC $CO_2$ flux  · $CO_2$ gradient based on three heights from ICOS (125 m, 67 m, 16 m) | · K (2)  · $CO_2$ or $O_2$ gradient based on two heights from campaign (125 m, 23 m) | · $CO_2$ flux (3)  · $O_2$ flux (3) | · $ER_{forest}$ (4)  · EC $CO_2$ flux  · GPP from ICOS database | - |

| (b) | Data used during the analysis of the aggregate day between 13 through 15 June 2019 [a] | | | | | |
|---|---|---|---|---|---|---|
| | (7) $ER_{atmos}$ | (8) Exchange coefficient (K) | (9) Surface flux of $O_2$ and $CO_2$ | (10) $ER_{forest}$ | (11) $ER_a$ and $ER_r$ | (12) GPP and TER |
| Data used: | - | · EC $CO_2$ flux  · $CO_2$ gradient based on three heights from ICOS (125 m, 67 m, 16 m) | · K (8)  · $CO_2$ or $O_2$ gradient based on two heights from campaign (125 m, 23 m) | · $CO_2$ flux (9)  · $O_2$ flux (9) | · $ER_a$ (7-12 July) (5)  · $ER_r$ (7-12 July) (5) | · $ER_{forest}$ (10)  · EC $CO_2$ flux  · $ER_a$ (5)  · $ER_r$ (5) |

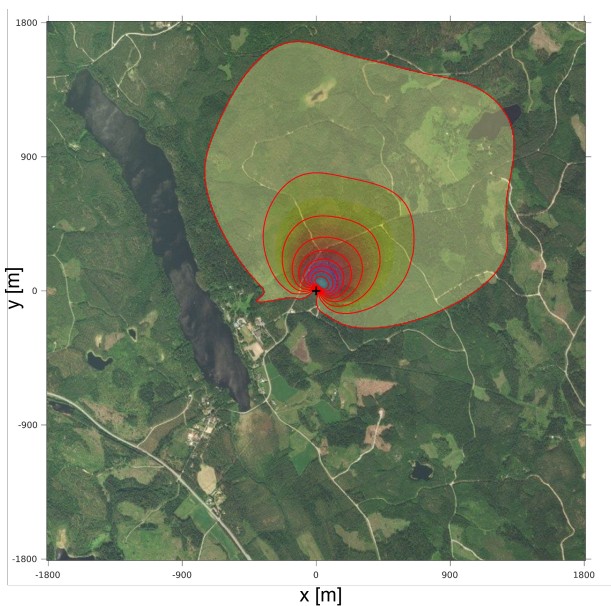

**Figure B2.** The footprint of the $O_2$ and $CO_2$ surface fluxes at 53.6 meters height (which is the geometric height), determined with the gradient method, for the days between 7 through 12 July 2019 at Hyytiälä. The lines and contours indicate the contributions to the footprint from 10% to 90% in steps of 10%. The plus sign (+) indicates the location of the tower. This figure was created with the method of Kljun et al. (2015)

*Author contributions.* ITL designed the measurement campaign and conducted the measurements. ITL, ERB, LNTN, PAP and ACM contributed to the design and development of the $O_2$ and $CO_2$ measurement setup. LNTN, BAMK, IM and TV contributed to the measurement campaigns. KAPF and ITL analyzed the measurements. KAPF, ITL, WP, JV, HAJM interpreted and discussed the methods and results. KAPF and ITL wrote the manuscript with input from all co-authors.

*Competing interests.* There are no competing interests.

*Acknowledgements.* The authors would like to thank Janne Levula (previously worked at Institute for Atmospheric and Earth System Research (INAR) / Physics, Faculty of Science, University of Helsinki, Helsinki, Finland) and Bert Heusinkveld (Meteorology and Air Quality, Wageningen University and Research, Wageningen, the Netherlands) for their help at Hyytiälä during the measurement campaigns, Marcel de Vries (Centre for Isotope Research (CIO), Energy and Sustainability Research Institute Groningen, University of Groningen, Groningen, the Netherlands) for technical support and Charlotte van Leeuwen (previously at Centre for Isotope Research (CIO), Energy and Sustainability Research Institute Groningen, University of Groningen, Groningen, the Netherlands) for the development of the instrument. This work was supported with funding that ITL received from the Netherlands Organisation for Scientific Research (016.Veni.171.095). We thank Üllar Rannik and Pasi Kolari (Institute for Atmospheric and Earth System Research (INAR) / Physics, Faculty of Science, University of Helsinki, Helsinki, Finland) for their work on the interpretation and analysis of the Hyytiälä flux measurements and their footprint. We thank the three anonymous reviewers for their comments which helped to improve this manuscript.

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
