# Peer review of "Diurnal variability of atmospheric O2, CO2 and their exchange ratio above a boreal forest in southern Finland"

_Atmospheric Chemistry and Physics, 2022_

## Author Response (AR1)

RC1: **'Comment on acp-2022-504'**, Anonymous Referee #1, 28 Aug 2022

**Review of "Diurnal variability of atmospheric $O_2$, $CO_2$ and their exchange ratio above a boreal forest in southern Finland" by Faassen et al.**

In this paper, the authors present diurnal variations in $\delta(O_2/N_2)$ and $CO_2$ observed at two heights above the boreal forest. They calculated $ER_{forest}$ and $ER_{atmos}$ based on flux and concentration measurements, respectively, and found $ER_{forest}$ and $ER_{atmos}$ cannot be used interchangeably. The authors also applied the observed $ER_{forest}$ to separate the NEE into GPP and TER, and they found comparable results to the commonly used eddy covariance approach. These findings supported and refined the discussion by Seibt et al. (2004) and Ishidoya et al. (2013, 2015) who reported differences between $ER_{forest}$ and $ER_{atmos}$ and its application to forest carbon cycle. There are only a few data sets of continuous measurements of both $\delta(O_2/N_2)$ and $CO_2$ over forests, and accurate estimate of $ER_{forest}$ at various forests is highly important for not only forest but also global carbon cycle. This paper makes a valuable contribution in this respect. However, I find some issues in the observed variations in $\delta(O_2/N_2)$ which should be addressed before publication.

We thank the reviewer for their review and assessment of our manuscript. We will address the issues on the observed variations and other points below. Note that line numbers given refer to the line numbers in preprint version.

Main Points

The authors ascribed the temporal decreases of $O_2$ and $CO_2$ between 13:00-20:00 (P3b) in Figure 4 to a remaining artefact that could not be corrected for with the pressure correction associated with the instability of the MKS pressure regulator in 2019. If so, I think the artefact also superimposed on the $O_2$ data during the other periods (P1, P2, and P3a), and I am concerned about the unrealistic values of $ER_{atmos}$ of 2.28±0.01 and 2.05±0.03 found in Fig. 5 are also attributed to the artefact. In my experience, larger $ER_{atmos}$ than 2.0 has never been observed in a diurnal cycle at a forest in a growing season. I recommend the authors to create the aggregate day based on the periods other than 7-13 July, 2019, and calculate the $ER_{atmos}$ for the average diurnal cycles. Especially, the $ER_{atmos}$ in 2018, when the pressure correction was not applied, will be useful for comparison. If larger $ER_{atmos}$ than 2.0 is also found in the average diurnal cycles in 2018, then the value will be reliable. However, if larger $ER_{atmos}$ than 2.0 is found only in the diurnal cycles in 2019, then it may be due to the artefact and the $ER_{forest}$ may also be affected by the artefact. To discuss differences between $ER_{forest}$ and $ER_{atmos}$ properly, it is important to rule out the possibility of the significant effect of the artefact.

We agree with the reviewer that the correction for the instability in the MKS pressure transducer is not sufficiently explained in the paper and that the effect of this correction on both the $ER_{forest}$ and the $ER_{atmos}$ signal can be explained more thoroughly. We address this as 3 different points: first, we give further details on the correction we made, secondly, we compare the 2018 $ER_{atmos}$ signals, and finally, we discuss the possible explanations for the $ER_{atmos}$ values higher than 2.0.

**1) Further details on MKS pressure transducer instability correction**

Figure 1 below shows how we determined the correction for the MKS pressure transducer instability. This figure shows the relationship between the four minute $\Delta(\Delta)O_2$ values of the three different calibration tanks (indicated each with a different symbol) per reference tank period (indicated by different colours) for 2018 and 2019 separately. The $\Delta(\Delta)O_2$ values are the deviations from the mean of each calibration tank. This figure clearly shows that the $\Delta(\Delta)O_2$ values in 2018 were not influenced by issues of the instability of the MKS pressure transducer. This is different for 2019, where the pressure difference of the MKS sensor shows a relationship with $\Delta(\Delta)O_2$. When the differential pressure values deviate further from zero, the $\Delta(\Delta)O_2$ values decrease. These pressure deviations are strongest for the period with the last used reference tank (#5) in 07-15 July 2019 and the aggregate day that we use in the paper is measured in this period. We derived the linear regression line for 2019, as shown in the figure, and used this to correct the 2019 $\Delta(\Delta)O_2$ measurements.

[Figure]

*Figure 1 The relationship between the differential pressure measured by the MKS sensor and the 4 minute $\Delta(\Delta)O2$ values of the calibration tank values during the 2018 and 2019 measurement period. The $\Delta(\Delta)O2$ values are the deviations from the mean of the specific calibration tank.*

[Figure]

*Figure 2 The corrected and not corrected 30-minute averaged O2 concentrations at 23 m for the PMKS deviations (a), together with the corrected and not corrected gradient of O2 (b). The time series for the 2019 measurement period of the PMKS (c) and the air temperature (d). Same as Figure B1 in the appendix.*

Figure 2 shows the impact of the correction on the measurements. This figure is now also included in the appendix of the paper. Figure 2a shows the effect of the pressure correction on the $O_2$ concentrations measured at 23 m, and the vertical gradient of $O_2$ during the 2019 measurement period. The figure shows that in the beginning of the measurement period, the pressure correction was minimal and at the end of the measurement campaign the correction increased, according to the increasing instability of the MKS pressure transducer. This corresponds to the data in Figure 1, where the period of reference tank 3, used in the period 07- 15 July, shows the strongest deviations in the PMKS. The shaded area indicates the period that we used to calculate the representative aggregate day.

The MKS pressure transducer instability correlated with air temperature. This is visible in panels c and d of figure 2.  With higher temperatures, the PMKS deviated further from zero compared to lower temperatures, although the malfunctioning of the pressure transducer prevented the differential pressure to go to zero completely, even at night. We therefore find different impacts of the correction between day and night, with a larger correction needed during the day compared to the night, increasing the uncertainty of the day-time measurements, compared to the night-time measurements. Based on this analysis, we therefore do not include the period 13:00-20:00 in our calculation of $ER_{atmos}$.

Compared to the concentration measurements, the gradient is hardly affected by the PMKS correction (Figure 2b).  Since the correction is applied at the two heights, the bias is cancelled in calculating the gradient as a difference between the two heights. $ER_{forest}$ is therefore not affected by this correction as it is based on the gradient, in contrast to $ER_{atmos}$ which is based on a single measurement height.

We have updated the text in lines 164, 320, 332 to further explain the correction.

**2) Comparison of $ER_{atmos}$ values between 2018 and 2019**

To make sure that the high values for $ER_{atmos}$ observed in 2019 are not caused by the pressure correction, we evaluated the $ER_{atmos}$ signals of several days in 2018. In 2018 no pressure correction was applied, because there was no instability in the MKS pressure transducer at that point (Figure 1). We selected several days in 2018 that showed a clear diurnal cycle for $CO_2$ and $O_2$ for which we calculated $ER_{atmos}$. We calculated $ER_{atmos}$ values for daytimes periods that include the entrainment process (between 5:00-13:00 LT). The obtained values of the daytime $ER_{atmos}$ ranged between 1.5 to 2.7, based on aggregates that have reasonable diurnal cycles and gradients of $O_2$. From these 2018 $ER_{atmos}$ values, we confirm that a value of 2.28 which we obtained for the aggregate day in 2019 is not uncommon and can be reproduced in a different year for a period in which pressure stabilization issues were absent.

**Explanations for the high $ER_{atmos}$ values**

Next to reviewer 1, also the other 2 reviewers had concerns about the high $ER_{atmos}$ values.  We will therefore elaborate here on the possible explanations for these values. Three possible explanations were given by the different reviewers:
- The PMKS correction could have caused these high numbers (all 3 reviewers). We show above that high $ER_{atmos}$ values were also found in 2018 when there were no issues with the MKS pressure stabilization.

- The lake could have affected the measurements (Reviewer #2). We analysed the footprint and the wind direction. In the answer to reviewer #2 we provide evidence that the dominant wind direction was between north and northeast. Therefore, the footprint of the concentration and flux measurements is not influenced by the lake and mainly influenced by boreal forest exchange.
- High $ER_{atmos}$ values should be linked to fossil fuel combustion (Reviewer #3). The main fossil fuel sources are to the south of the measurement tower. As the main wind direction is from the north to northeast throughout the duration of the aggregate day, it is unlikely that fossil fuel combustion strongly influenced our signal.

These possibilities mentioned by the reviewers do not explain the high $ER_{atmos}$ signals, but we do agree with the reviewers that such high values for $ER_{atmos}$ of 2.05 for the entire day and of 2.28 for the period P2 are indeed questionable in the context of previous values reported in literature, based on ecophysiological relationships between respiration and photosynthesis.

Although we cannot fully rule out remaining artefacts in the calibration due to e.g. temperature changes in the measurement cabin, we suggest that the more plausible explanation is that $ER_{atmos}$ is highly influenced by atmospheric processes. These processes are closely linked to the development of the atmospheric boundary layer dynamics.

We are currently working on extending the work presented in this manuscript with a modelling study, to support the data analysis. We use the Chemistry Land-surface Atmosphere Soil Slab (CLASS) model (Vilà-Guerau de Arellano et al., 2015) and have implemented the $O_2$ and $CO_2$ exchange. We are therefore able to disentangle which individual contributions determine $ER_{atmos}$ and how such high values could have formed. Our preliminary results show that entrainment of air masses originated at the residual layer, with different characteristics of $O_2$ and $CO_2$, leading to the dominant contribution after sunrise (figure 3 between 04:00 – 09:00). During this period the signal of the measured $O_2$ and $CO_2$ concentration is more determine by atmospheric factors than by surface factors.

[Figure]

*Figure 1 The change of O₂ and CO₂ over time, split into the different processes that contribute to the total signal, modelled with CLASS for the representative aggregate day (7-12 July). The shaded colours indicate the same time periods as indicated in Figure 4a in the paper.*

The high values that we find for $ER_{atmos}$ can therefore be explained by the entrainment process, and are influenced by a difference between $CO_2$ and $O_2$ in what we call the jump. The jump is the difference between the concentration inside the boundary layer compared to the free troposphere or the residual layer. A difference in the jump of $O_2$ and $CO_2$ can arise when different sources of background air are mixed. For example, ocean $CO_2$ and $O_2$ exchange are decoupled, so when an ocean $CO_2$ sink is present in the background signal, the air will be depleted of $CO_2$ whereas the $O_2$ concentration will be hardly affected. When sources of air mix with different ER signals, we cannot just simply average these ER signals as they could describe different states of the atmosphere (see line 570, or point 3 below). This would mean that the resulting mixture of $O_2$ and $CO_2$ could have an ER that is an 'unrealistic' value when we compared it to ER signals of single processes. The air in the troposphere is a mixture of air from different background sources. Our preliminary model results show (not shown here) that even a slight difference in jump between $O_2$ and $CO_2$ would result in a change in $ER_{atmos}$. This confirms our message that the $ER_{atmos}$ signal is sensitive to entrainment, and that we should always look at $ER_{forest}$ if we would like to indicate the surface ER. If no entrainment would occur, $ER_{atmos}$ would be the same as $ER_{forest}$. $ER_{atmos}$ could therefore be used to help quantify the relevance of the biosphere processes with respect to atmospheric driven processes.

This explanation of the entrainment processes dominating the values we find for $ER_{atmos}$ does not directly explain why we get such high $ER_{atmos}$ values compared to other studies. We think this can be explained to the measurement height. The studies of Seibt et al. (2004) and Battle et al., 2019 measure closer to the canopy compared to our study. The effect of entrainment on the measurements is less when measuring closer to or even inside the canopy. When looking at the $ER_{atmos}$ values of the 125 m from our study we find an $ER_{atmos}$ signal for P2 of 3.40. This number is even higher compared to the 23 meters value of 2.28. For the study of Ishidoya et al. (2015) it is not completely clear to us how high they measured compared to the canopy as they assume a range in canopy height. It seems that their highest measurement height is at a similar distance to the canopy compared to our lowest height (23 m). However, they find only a small difference between $ER_{atmos}$ and $ER_{forest}$. A possible explanation for this could be the difference in background air that is entrained from the free troposphere or that surface exchange dominated over the atmospheric effects. Based on the different findings between these studies we suggest that more research is needed, especially focussing on measuring the

O$_2$ and CO$_2$ concentration also in the free troposphere and further modelling studies to better determine the drivers of O$_2$ and CO$_2$ by biosphere or atmosphere effects.

We have changed the text in line 481 to better clarify our explanation of the high values of ER$_{atmos}$ linked to entrainment.

Other Specific Points

1) Line 175-178 and Table 1: What does "our own calibration" mean? Did the authors calibrate the target cylinder using the primary Scripps cylinders by themselves? I think the declared value with calibration in Groningen is based on SIO scale. Therefore, the values of target cylinder based on "our own calibration" should also be on SIO scale to calculate the mean of the difference.

We agree with the reviewer that this is unclear. We did not mean to say that we have used two different calibrations, what we meant was indeed that we use the values calibrated at Groningen using the SIO cylinders there. We have updated the text accordingly.

2) Line 190-192: Related to the main points, the period of 7 through 12 July 2019 to create the aggregate day is shorter than that by Ishidoya et al. (2015). I am concerned about the artefact during this period considering the very high ER$_{atmos}$ found in Fig. 5.

We agree with the reviewer that we used indeed less days compared to the study by Ishidoya et al. (2015) and that by adding more days we could have made our results more robust. However, the reasons we used only these six days was because before and after this period the measurements were not showing clear diurnal cycles and gradients. We also wanted to use consecutive days and because our measurement period was relatively short, we used only these six days. These days were chosen based on their gradient of O$_2$ and the meteorological conditions, as is describe in lines 182-185. Furthermore, our measurements in 2018 also give high values for ER$_{atmos}$, which shows that the high values are not dependent on the artefact of the PMKS correction, but rather are explained by entrainment which we have further elaborated above in reply to the first major point.

3) Line 223-226: The authors calculated the ER$_{forest}$ from means of the O$_2$ and CO$_2$ flux during night, day, and entire day. I think it can also be calculated by applying a linear regression between O$_2$ and CO$_2$ flux (or $\Delta$O$_2$ and $\Delta$CO$_2$) on the points as Ishidoya et al. (2015, 2020) did. Wouldn't this method reduce the uncertainty on ER$_{forest}$?

It is indeed not obvious from previous literature how to correctly infer ER$_{forest}$ from the O$_2$ and CO$_2$ flux data. We explain here how we arrived to our approach, after considering 1) the slopes of the gradients and 2) the slopes of the fluxes.

**1) Slope of the O2-CO2 vertical gradients**

We first considered the methods that Ishidoya et al. (2015, 2020) use, which derives ER$_{forest}$ from the gradients alone. In figure 3a below the vertical gradient of O$_2$ is shown against the gradient of CO$_2$. The slope of a linear regression through all the points (24 hours) is -1.028.

[Figure]

*Figure 2 the vertical gradient of $O_2$ against the vertical gradient of $CO_2$ (a), together with the surface flux of $O_2$ against the surface flux of $CO_2$ (b) of the aggregate day. The shaded blue part is the period that indicates the night and the shaded red is the period that indicates the day.*

However, by looking only at the gradients, the size of the flux itself is not taken into account. The size of the flux is calculated as a combination of the vertical gradients and the turbulence of the atmosphere is (K in equation 6). To get the correct ER values that represent the complete forest, the fluxes need to be weighted according to their contributions.

For example, during the night, the gradient is relatively large, and changes rapidly (blue area in Figure 3a) compared to the daytime (red area in Figure 3a). These large gradients during the night, are mainly caused by strengthening of the thermal stratification and somewhat by the surface fluxes. If we would focus on the gradient changes, we would therefore focus on the periods where the stability of the atmosphere changes fast and not focus on the periods where the surface fluxes are the largest (during mid-day). When determining the ER$_{forest}$ of an ecosystem it is important to take into account the size of the flux as the largest flux contributes the most to the final ER$_{forest}$. However, during the day, the gradients are very small due to the more active turbulent mixing, while the fluxes are larger than during the night. Also, the transition periods (from gradient-dominant stable nocturnal conditions to flux-dominant convective diurnal conditions) takes place during the day. During these transitional periods it is recommended to be careful in using the vertical gradient.

During the night, the surface fluxes are relatively constant (Figure 6 in paper), and the slope of the gradients as in figure 3a would give a value of -1.028, which is close to the ER$_{forest}$ value we derived for the night with the methods described in our paper. Applying this method during the night would not per se reduce our uncertainty, as our surface fluxes are already quite certain during the night.

We added additional information in lines 226 and 559 to further explain this, and to better highlight the difference to the methods of Ishidoya et al. (2015).

**2) Slopes of the $O_2$-$CO_2$ fluxes**

The other option is to apply a linear regression to the $O_2$ and $CO_2$ fluxes, as was suggested by the reviewer (Figure 3b). However, this approach also has three issues that we discuss here below, showing that this approach does not necessarily reduce the uncertainty.

First, we cannot combine the day and night measurements and calculate one average from all the values. This is because the surface fluxes of $O_2$ and $CO_2$ change sign between the night and day because the processes that happen, both biosphere and atmospheric, are different. By changing sign, the ER for the day and the night indicate different processes by which the surface is affecting the atmosphere. Where for example during the night, the surface fluxes deplete atmospheric $O_2$ and during the day increases atmospheric $O_2$. We therefore cannot average the ER signals. This also shows in the $ER_{forest}$ values in table 3, where the $ER_{forest}$ value of 24 hours is not the average of the day and the night value. This is the reason why we first average the surface fluxes over the entire day and then use these averaged surface fluxes to calculate the $ER_{forest}$.

Second, the linear regression between the $O_2$ and $CO_2$ flux should give a correct ER when there only one process active, which is the case during the night (blue area in figure 3b). However, during the night the surface fluxes are quite constant in time, and it is therefore difficult to derive the linear regression, as most data points are close together. The derived slope (and resulting ER) becomes then sensitive to outliers.

Third, when two processes occur at the same time (during the day: red area in figure 3b), a linear regression would result in an average ER for the period we selected but ignores the fact that there are time steps with larger fluxes and thus should have a larger impact on the final ER. This is the same problem as described for the gradients above. Linear regressions for the day-time period (09:00-17:00) for the gradients (figure 3a) and the fluxes (figure 3b) would roughly give the same value for the slopes: -1.85 and -1.87 respectively. However, this should not be the case, as we showed above that the gradients and fluxes do not represent the same information.

Based on these arguments, we have chosen our method as described in the paper to calculate the $ER_{forest}$ based on the average of the fluxes, giving, giving a weighted value of the $ER_{forest}$ signal. We have elaborated more on this point and therefore added some text in lines 398 + 552 + 559.

4) Figure 4: Do the error bars indicate standard error? Please specify.

We agree with the reviewer that it is unclear throughout the text how exactly we determine the error bars in the figures and the resulting uncertainty values of the different ER signals. We therefore added extra information to the methods section that explains how we determined the error bars and the uncertainties of our data, see lines 166 + 192, and the added equation 4 after line 192.

We also added this extra information to the caption of figure 4 on how the error bars in figure 4b are determined, together with extra text in lines 332, 380 and 401.

5) Figure 7: The $ER_{forest}$ is negative value in this figure, although it is defined as positive value throughout the paper. Please be consistent with the terms you use.

We agree with the reviewer that we were indeed not completely consistent with the negative or positive values of the $ER_{forest}$ signals. We therefore changed figure 7, and updated the text accordingly where necessary.

6) I think it would be better to add the references and/or brief description of the EC method and temperature-based function used in this study, since comparison of EC method and O₂ method in Fig. 8 is an important topic.

We agree with the reviewer that we should elaborate a bit further how the data that we use from ICOS and how they approached the EC method. We therefore added the reference of Kulmala et al. (2019) and more text in lines 271.

7) The words "Eddy Covariance (EC)" appears repeatedly at line 30, 131, 227, and "Eddy Covariance fluxes" and "eddy-covariance CO₂ flux" also appear at line 429 and 634, respectively. I think it's better to use "EC" throughout the paper after the definition at line 30.

We agree with the reviewer that we can use only the term EC after introducing it in the introduction and do not have to write it completely out anymore after that. Therefore, we made sure that throughout the paper only EC is used to refer to the eddy covariance technique.

**Citation**: https://doi.org/10.5194/acp-2022-504-RC1

References

Battle, M. O., William Munger, J., Conley, M., Sofen, E., Perry, R., Hart, R., Davis, Z., Scheckman, J., Woogerd, J., Graeter, K., Seekins, S., David, S., & Carpenter, J. (2019). Atmospheric measurements of the terrestrial O2 : CO2 exchange ratio of a midlatitude forest. *Atmospheric Chemistry and Physics*, *19*(13), 8687–8701. https://doi.org/10.5194/acp-19-8687-2019

Ishidoya, S., Murayama, S., Kondo, H., Saigusa, N., Kishimoto-Mo, A. W., & Yamamoto, S. (2015). Observation of O2:CO2 exchange ratio for net turbulent fluxes and its application to forest carbon cycles. *Ecological Research*, *30*(2), 225–234. https://doi.org/10.1007/s11284-014-1241-3

Ishidoya, S., Sugawara, H., Terao, Y., Kaneyasu, N., Aoki, N., Tsuboi, K., & Kondo, H. (2020). O2:CO2 exchange ratio for net turbulent flux observed in an urban area of Tokyo, Japan, and its application to an evaluation of anthropogenic CO2 emissions. *Atmospheric Chemistry and Physics*, *20*(9), 5293–5308. https://doi.org/10.5194/acp-20-5293-2020

Kulmala, L., Pumpanen, J., Kolari, P., Dengel, S., Berninger, F., Köster, K., Matkala, L., Vanhatalo, A., Vesala, T., & Bäck, J. (2019). Inter- and intra-annual dynamics of photosynthesis differ between forest floor vegetation and tree canopy in a subarctic Scots pine stand. *Agricultural and Forest Meteorology*, *271*(February), 1–11. https://doi.org/10.1016/j.agrformet.2019.02.029

Seibt, U., Brand, W. A., Heimann, M., Lloyd, J., Severinghaus, J. P., & Wingate, L. (2004). Observations of O2: CO2 exchange ratios during ecosystem gas exchange. *Global Biogeochemical Cycles*, *18*(4), 1–18. https://doi.org/10.1029/2004GB002242

Vilà-Guerau de Arellano, J., van Heerwaarden, C. C., van Stratum, B. J. H., & van den Dries, K. (2015). *Atmospheric Boundary Layer: Integrating Air Chemistry and Land Interactions*. Cambridge University Press. https://doi.org/DOI: 10.1017/CBO9781316117422

**RC2: 'Comment on acp-2022-504', Anonymous Referee #2, 06 Sep 2022**

**Diurnal variability of atmospheric $O_2$, $CO_2$ and their exchange ratio above a boreal forest in southern Finland**

Faassen et al. present a highly novel dataset of $O_2$ and $CO_2$ measurement in the surface layer over a boreal forest. Such measurements are technically very challenging making this study one of the very few so far that have succeeded to apply $O_2$ in micrometeorological land surface flux measurements. Typically, the signal to noise ratio in $O_2$ gradient above forests is very small which limits the application of the flux gradient methods. Here the authors make use of a 125 m tall tower to increase the $O_2$ gradient.

A major challenge in this study is that the measurement uncertainty of the $O_2$ system is below comparable systems. This limits the interpretation of the data. Nevertheless, in my view the authors found a suitable way forward by aggregating the data to a "representative day".

While the experimental design and analysis is well done, there are several aspects that need to be addressed before publication.

We thank the reviewer for their review and assessment of our manuscript. We will address the points raised below. Note that line numbers given refer to the line numbers in preprint version.

Major comments

- Footprints: A major question regarding the study is that the ERatmos values are much higher than in previous studies. Some potential reasons are discussed in lines 481 to 490. What I am, however, missing is a proper treatment of the concentration footprints. Firstly, they differ between heights, particularly if the height difference is 100 m. This could lead to situations where the bottom height sees the local land surface whereas the top height sees air influenced at a regional level. Secondly, right next to the towers (roughly 200 m) is a large lake. Given that lakes have different $O_2$:$CO_2$ exchange ratios, I am wondering how this would influence the observed signal. Some of the co-autors have published articles on eddy covariance flux measurements over that lake. For the manuscript it would be help to add a footprint analysis and evaluate and discuss the influence of these two aspects on ERatmos and ERforest.

  For a detailed explanation on the high values of ERatmos compared to previous studies, we would like to refer to our answer of point 1 of reviewer #1. We explain there why we think entrainment is the most likely explanation of the high $ER_{atmos}$ values that we observed.

  We agree with the reviewer that a more extended explanation about the footprints of the measurements should be added. We have therefore calculated the footprint for the flux gradient (Figure 1) and we looked in more detail at the footprint of the concentration measurements.

  Looking in more detail at the footprint of the concentrations measured at 23 m and 125 m (Carbon Portal ICOS RI, 2022), we find that the measurements at both heights

are strongly characterized by background signals, mainly forest and also ocean. The influence of fossil fuel is limited at both heights. Despite the high influence of background signals on both measurement heights, both heights are still able to capture the diurnal cycle of the forest. Our measurements heights are not disconnected from the surface but they integrate the surface and the atmosphere, as we observe two clear diurnal cycles. This suggests that both heights are measuring roughly the same areas. We agree with the reviewer that the difference between the heights (100 m) is quite large compared to previous studies. This large difference was used to be sure we were able to measure a $O_2$ gradient. If we can improve the accuracy of our $O_2$ measurements, we would certainly choose a smaller difference between the two measurement heights for a next campaign or add a third measurement height.

[Figure]

*Figure 1 The footprint of the O2 and CO2 surface flux, determine with the gradient method, for the days between 7 through 12 July 2019 at Hyytiälä. The lines and contours indicate the contributions to the footprint from 10% to 90% in steps of 10%. The plus sign (+) indicates the place of the tower. The lake is located west of the measurement tower without influencing the measurements. This is the same Figure as figure B3 in the paper.*

For the flux calculations based on the gradient method we calculated the footprint with the method of Kljun et al. (2015) by using the geometric height (Figure 1). Figure 1 shows that the main wind direction during the aggregate day was coming from North to Northeast and therefore the lake did not fell inside the footprint for these days.

To be clearer about the footprints of the $ER_{atmos}$ and $ER_{forest}$ signals we modified the text in the line 289 and added Figure 1 to the appendix.

- Flux partitioning: It I understood correctly, the exchange ratio of assimilation (ERa) is calculated based on equation 8 assuming a constant ERr and ICOS data of NEE, GPP and TER (line 276). Once ERa and ERr are retrieved for one representative day, these values are used to calculate GPP and TER on other days. For me it is not clear what we learn from this exercise as GPP is used to constrain ERa and then ERa used to constrain

GPP. Other studies such as Wehr et al. 2016 Nature have shown that NEE partitioning with an independent method using 13C in $CO_2$ resulted in lower TER and lower GPP compared to the temperature-based function following Reichstein et al. 2005 possibly indicating a Kok effect. If now the temperature-based GPP is used to calculate ERa, the $O_2$ based method does not provide additional and independent information. While I understand that the authors have no independent measurements of ERa at hand, I still miss a more careful discussion including Wehr et al. 2016 and addressing the limits of this approach.

We agree with the reviewer that our method is indeed not completely independent when estimating GPP and TER with the $O_2$ method. We indeed use GPP to estimate ERa and use this ERa on another set of days to estimate GPP again. However, this analysis is not completely circular, as we calculate a new $ER_{forest}$ for a new set of days. However, due to this assumption, we could not make a completely independent comparison between the EC method and the $O_2$ method, but we show this to give a first estimate of the flux partitioning using $O_2$ to highlight the benefits of using the $O_2$ method. Further research will indeed focus on separately deriving ERa independently from EC.

Our goal with the analysis presented in Figure 8, was to determine if the $O_2$ method gave similar results compared to the EC method, by applying the ER values we determine during this study to another day. In the text we discuss that we find realistic values for ERa and ERr and therefore we are confident that they could be used in the $O_2$ method. Independent estimates of ERa would likely result in only minor changes compared to our estimate. Small changes in ERa would have similar effect on the uncertainty of the partitioning compared to small changes in ERforest (Figure 8). The biggest source of variability in the $O_2$ method is the ERforest of the new set of days. The ERforest depends on the $O_2$ gradient, and this gradient was difficult to determine during our measurement campaign because of the relatively low measurement precision. However, figure 8 still shows that we find a good comparison between the O2 and the EC method for the partitioning of the fluxes and we present it to show the potential of the method.

To independently determined ERa and ERr, one would use branch and soil chambers, or lab measurements to measure process-level $O_2$ and $CO_2$ exchange which were not included unfortunately in our campaigns. When chamber measurements would have been available, we could have made a more detailed comparison between the $O_2$ and EC method and compare this result with the study of Wehr et al. (2016). With a more detailed analysis we could potentially say of the $O_2$ method gives a higher or lower estimation of GPP and TER and if this differs compared to the 13C method from Wehr et al. (2016).

We added text to line 611 to make to address these points.

Minor comments

- Line 23: better "net uptake" than "uptake":

We agree with the reviewer and changed therefore uptake into net uptake in line 23.

- Line 23 and 24: better be consistent using ether land biosphere or terrestrial biosphere:

We agree with the reviewer on this point and changed therefore land biosphere into terrestrial biosphere in the lines: 23, 44, 45.

- Line 27: Add a citation for last sentence in first paragraph.

We agree with the reviewer and have added citations the end of the first paragraph in line 27.

- Line 30-32: here I am missing a mentioning of Wehr R et al. 2016 Nature where they showed that fluxes partitioning using 13C differ from fluxes partitioning following Reichstein et al. 2005.

We agree with the reviewer and have added the citation of Wehr et al. (2016) to line 32. For a more elaborate comparison between our results and Wehr et al. (2016), please look at the discussion above of the second major point.

- Line 36/7: fluxes of $O_2$ and $CO_2$ are opposite. Here a positive ER is used. It might be helpful to indicate this by saying "indicates the amount of moles O2 consumed per mole of CO2 produced (or vice versa)".

We agree with the reviewer that this sentence should be made clearer. Therefore, we changed line 36/37.

- Figure 1: in the text of the introduction the term GPP and TER are used and in figure 1 respiration and assimilation. Please use consistent terms.

We agree with the reviewer that we have to be consistent when using these terms and have to be clear what we mean with them. Therefore, we added Gross Primary Production (GPP) and Total Ecosystem Respiration (TER) in figure 1b, to make the link with the text clearer. However, our opinion is that we cannot use GPP and assimilation interchangeably because they indicate different scales. GPP is a measurement that applies to the ecosystem level and assimilation is more related to the process/leaf level. To make this more clear, we changed the text in the caption of Figure 1.

- Line 90 to 94: personally, I prefer if the given objectives are presented with the term "objectives" for allowing speed-reading. Maybe a matter of taste

We added the word 'objective' in line 90 to make it easier for people that want to speed-read this paper.

- Line 113: what is the influence of the nearby lake on the exchange ratio. The footprints at 23 m and at 125 m are very different. How does this influence the results?

The lake has minimal influence on the measurements both at 23 m and 125 m, as they both mainly measure background signals. The difference in footprint becomes important when looking at the surface flux calculations. The footprint of the flux calculations shows that the signal we measured did not originate from the lake during our representative day (Figure 1). Air masses that are influenced by the presence of the lake are therefore absent. We elaborated more on the footprints at the first major point in this review above, and have changes the text at line 289 and added Figure 1 above to the appendix.

- Line 129/130: It seems that the sampling lines are alternatingly flushed with 120 ml/min and 2 l/min. Has it been evaluated whether these changes in flow rate lead to any effects on the $O_2$ signal? Or are all these effects removed by discarding the first 4 minutes after switching.

These effects are indeed removed by discarding the first 4 minutes after switching heights, as the flow rates adapt within that time. We use these different flow rates to make sure the samples lines are properly flushed before switching, preventing delays in the measurements of the new height.

- Line 210: I find it confusing that in equation 5, eddy covariance terms for the turbulent fluxes are presented, but the turbulent fluxes are obtained from flux gradient measurement. Why are not equation 5 and 6 combined?

We agree with the reviewer that it can be confusing to use the eddy covariance terms, while we calculate the surface fluxes of $O_2$ and $CO_2$ with the gradient method. Therefore, we changed the terms that indicate the surface fluxes from $w'CO_2'$ and $w'O_2'$ into $F_{CO2}$ and $F_{O2}$ throughout the paper to clearly show that they are inferred from gradients.

We did not want to combine equation 5 and 6, to make it as clear as possible how $ER_{forest}$ should be calculated from the surface fluxes of $O_2$ and $CO_2$. These could (in other setups) possibly also be measured directly from EC measurements of $O_2$ and $CO_2$. However, we derived these from the vertical gradients and therefore showed in a separate equation how we did that. To make the connection between equation 5 and 6 clearer we now refer to equation 6 in line 215. (Note that an equation was added (equation 4) in the new version of the paper, and therefore the numbers of the equations in the paper changed).

- Line 218: In my view it is not the stability that characterised if in a period respiration or assimilation dominate, but it is the radiation regime. Why was here stability used and not nighttime vs. daytime?

We agree with the reviewer that indeed the radiation regime is also of importance to distinguish between the different periods. To make it more clear in the text we changed line 218.

- Line 255: unit is missing. Should be "0.4 m s-1".

We agree with the reviewer about this point and changed therefore the text in line 255.

- Line 271: here it is referred to ICOS NEE and GPP from EC measurements. It would be good to say how ICOS partitions NEE into GPP.

We agree with the reviewer and have added a reference and more information in line 275 on how the GPP is determined in Hyytiälä.

- Line 304: why was a fixed calibration time during the day selected (20:00 - 22:00). An alternative could be using a moving calibration time.

We agree with the reviewer that a moving calibration time would benefit our data, and we generally applied a moving calibration time period for both the 2018 and 2019 campaigns. The calibration tanks are measured every 23 hours (see line 169) and take 2 hours to

complete. However, between the 7th and 13th of July we decided to fix the calibration time. During this period radiosondes were launched at Hyytiälä, which we will use for a follow up study. To make sure we captured the morning transition well during this period, we decided to fix the calibration time between 20:00-22:00, to allow a smoother planning of the radiosonde launches. At the time of this decision, we did not know we wanted to make an aggregate representative day. By coincidence the period between 7th through 12th July resulted in the best $O_2$ gradients and therefore this fixed calibration time is present in our representative aggregate day.

To make this more clear we adjusted the text in line 304.

- Line 307: 0.70 ± 0.65: the unit is missing.

We agree with the reviewer about this point and changed therefore the text in line 307.

- Fig. 4a: for the height 23 m, the $CO_2$ concentration varies with a range of 15 ppm, whereas the $O_2$ concentration varies with a range of 35 ppmEq. Wouldn't we expect to see a similar range of variation? What is the role of the nearby lake?

If our signal was only influenced by the forest, we would indeed expect a similar range for both $O_2$ and $CO_2$. However, the concentration measurements are highly influenced by entrainment as well. How much entrainment is influencing the data depends on the boundary layer growth and the difference in concentration between the boundary layer and the free troposphere, also called the jump. If there is a difference in this jump for $O_2$ and $CO_2$, this will result in different range of the effect of entrainment. It is highly likely that the jump of $O_2$ and $CO_2$ are not the same, as the background air in the free troposphere contains different sources of e.g. fossil fuel emissions and ocean exchange. These processes affect the $O_2$ and $CO_2$ concentration differently. The air that is entrained could also be affected by the memory of the day before or advection during the night inside the residual layer. A more elaborate explanation of how we think entrainment influenced our measurements is given in the first answer to reviewer #1 (see 'Explanations for high $ER_{atmos}$ values'), and we have update the text in line 481 to further explain this..

The lake will have had likely very little effect on our measurements, as was already discussed with the first major point, see above.

- Fig. 4b: at night we see a vertical gradient in $O_2$ concentration (roughly 10 ppmEq) that exceed instrument precision (roughly 4 ppmEq), but during daytime the gradient is – even averaged over multiple days – lower than instrument precision. To me it is unclear how the uncertainty of the measurements is propagated to the final fluxes and ERforest.

We agree with the reviewer that it was not clearly described how the uncertainties on our measurements were calculated. The error bars in figure 4b are not the same as the instrument precision presented in table 1. The error bars in figure 4b, the following figures, and the uncertainties in the ER signals are all based on the standard error of the 30-minute average $CO_2$ and $O_2$ concentrations. The error bars in 4b are then determined by calculating the uncertainty for the aggregate data points, based on this standard error (that we included in the revised version of the paper as equation 4) and by using error propagation. To make this more clear, we added additional information to the text, in lines 166 + 192, and the added

equation 4 after line 192. We also added this information to the caption of figure 4, together with extra text in lines 332, 380 and 401. See also point 4 of reviewer #1.

- Line 318: in P3b: $O_2$ and $CO_2$ concentration changes show the same sign, instead of the expected opposite sign. This is related to an instability of the MKS pressure regulator. It is unclear why this effect should only affect P3b and no other times of the day. How was this evaluated?

We agree with the reviewer that this was not clearly explained in the text. We give an elaborate explanation about this in the response of the first major point of reviewer #1 and added a new figure to the appendix. Please see our response to reviewer #1 for our answer and the updates to the paper.

- Fig. 5: Which regression type was used to calculate the regression?

This was a linear regression. To clarify this, we changed the text in line 208 and the caption of figure 5.

- Line 339: Given that the measurement uncertainty is so high compared to the variation during P3a, I am wondering how the uncertainty could be included via error propagation when calculated the slope and its uncertainty.

The measurement uncertainty per data point of the $O_2$ concentrations of the aggregate is around 2 ppmEq (see Figure 4). This is relatively low compared to the increase of $O_2$ during the day (30 ppmEq) or the decrease during the night (15 ppm Eq). Incorporating the uncertainty of the datapoints into the slope has therefore little effect on the uncertainty of the slope.

The largest uncertainty in the $ER_{atmos}$ values is the definition of the time boundary of where the influence of entrainment stops, which is difficult to incorporate into the slope of the linear regression. We therefore leave the uncertainty of the $ER_{atmos}$ values as it is, and we indicate that the $ER_{atmos}$ value of P3a could deviate a lot when the time boundaries of this period shifts.

- Fig. 6: The units of the fluxes are given in ppm m m-1. This is very unusual for the flux community. Typically, the fluxes are reported in μmol m-2 s-1. Also, I find it confusing that the y-axis label is the covariance, but the fluxes are calculated from a flux-gradient approach and not from eddy covariance.

We agree with the reviewer about these points and changed therefore the units of the fluxes to μmol m-2 s-1 in the text and the figures, and removed the eddy covariance notations from the paper. As mentioned before, we now stated clearly that the fluxes are inferred from gradients using flux-gradient method formulations.

- Fig. 6b: Could please describe in the caption what are the error bars. Could just be moved from the main text (line 380). Also here, it is unclear to me if an error propagation incl. measurement uncertainty was carried out.

We agree with the reviewer about this point and changed the caption of figure 6. We also clarified the calculations of the error bars in the text, see our answer above and our answer to reviewer #1 for more explanation.

- Fig. 7 : It is surprising to see ERforest values at -2 to -2.5. This is much more negative that other reported data and it is unclear what this could mean physiologically. It is also surprising that the fluxes with the most negative values are also the largest fluxes, where we would expect to see large gradients and thus robust flux calculations.

We agree with the reviewer that $ER_{forest}$ values of 2 to 2.5 are highly unlikely if they would only represent forest exchange. However, we do not think data points in figure 7 can be interpreted individually. As was explained in the manuscript, the concentration measurements have a relatively high uncertainty compared to previous studies and therefore the vertical gradient is difficult to determine. We therefore use an aggregate day to get a more robust estimate for the $O_2$ gradient, however with a remaining uncertainty on the individual data points. This also translate into the individual estimates for $ER_{forest}$ per time step, where for some time steps, very high $O_2$ surface fluxes caused such unrealistic values for $ER_{forest}$. We therefore use the average $O_2$ and $CO_2$ surface fluxes to determine the final $ER_{forest}$ values.

As the reviewer states, the large $O_2$ surface fluxes during the day follow from a larger vertical gradient of $O_2$. However, this gradient is still small compared to the gradient during the night and therefore still had a relatively large uncertainty. A large gradient during the day could therefore result from the relatively large measurement uncertainty, and as a consequence result in these high $O_2$ fluxes and unrealistic $ER_{forest}$ values.

- Appendix: Personally, I prefer that the units are shown as well.

We agree with the reviewer about this point and therefore we have added units to the equations in the appendix.

**Citation**: https://doi.org/10.5194/acp-2022-504-RC2

References:

Carbon Portal ICOS RI. (2022). *STILT station characterization for Hyytiälä at 17m*. Carbon Portal.
Kljun, N., Calanca, P., Rotach, M. W., & Schmid, H. P. (2015). A simple two-dimensional parameterisation for Flux Footprint Prediction (FFP). *Geoscientific Model Development*, *8*(11). https://doi.org/10.5194/gmd-8-3695-2015
Wehr, R., Munger, J. W., McManus, J. B., Nelson, D. D., Zahniser, M. S., Davidson, E. A., Wofsy, S. C., & Saleska, S. R. (2016). Seasonality of temperate forest photosynthesis and daytime respiration. *Nature*, *534*(7609). https://doi.org/10.1038/nature17966

RC3: 'Comment on acp-2022-504', Anonymous Referee #3, 22 Sep 2022

**Diurnal variability of atmospheric $O_2$, $CO_2$ and their exchange ratio above a boreal forest in southern Finland**

Overall, this is very nice paper presenting important results. The authors conduct challenging measurements, analyze the data intelligently and combine their own data with ancillary datasets in a clever way to extract interesting values. They focus on the O2/CO2 exchange ratio for a boreal forest (unprecedented) and then extend their work to seperately assess the exchange ratios associated with respiration and assimilation. The work is valuable, the paper is generally well organized and it definitely deserves publication.

We thank this reviewer for their assessment of our manuscript, including the detailed comments on the text. We will address remaining issues below.

That said, I do have some concerns that need to be addressed prior to full acceptance:

1. The authors use $\alpha_b$, ER and OR somewhat interchangeably in the introductory part of the paper. Each of these symbols really does have a distinct and specific meaning. Although the use of these terms in the literature has been somewhat sloppy, as our field matures it becomes more important to use the right word in the right context.

   We agree with the reviewer that in our field the terms alpha_b, ER and OR are used somewhat interchangeably in the literature and that we should be careful what we mean with each of them. We tried to use the terms correctly throughout our manuscript, but we agree with the reviewer that in some parts of the text we did not succeed in making a clear distinction between the different terms. Therefore, we changed the introduction text to be more consistent. The exact changes, with respective line numbers can be found below, where we have included the list with notes from the pdf, and a reply to these specific points one by one.

2. The authors assume ERr is constant day and night. This may well be true, but it's possible it isn't true. Since this assumption is central to the subsequent analysis, there should be more discussion of this assumption and its validity.

   We agree with the reviewer that more discussion should be added to strengthen the argument that the ERr can be assumed to be constant during the day and night. Not many studies have researched the changes in ecosystem ERr and ERa.

   Hilman et al. (2022) showed that the major contributor to the changes in ERr is the bulk soil respiration (the bulk soil is the part of the soil that is not influenced by roots). The bulk soil respiration changes with soil temperature and soil moisture. These changes are not likely to affect our diurnal cycle of ERr and mainly show effect on seasonal time scales. One component that could potentially change ERr during the diurnal cycle is the respiration of plants. As the respiration of the plants involves photorespiration during daytime and only dark respiration during nighttime. Dark- and photorespiration use different pathways and therefore may have a different ERr values and could affect the ERr of the ecosystem. To our knowledge no studies so far have looked in detail at the diurnal cycle of ERr of plant respiration and it is therefore difficult to say how much effect this has on the total ecosystem ERr. However, Hilman et al. (2022) showed that the

bulk soil respiration has the largest effect on the variability of ERr, so we assume that the variability of plant respiration has little effect on the total ERr of the ecosystem. Other studies that looked at the ERr also only focussed on the soil and only focussed on a longer time scale compared to the daily cycle (Angert et al., 2015; Pries et al., 2020). Therefore, more research should be performed to check if our assumption is valid.

The ERa seems to be influenced by the source of nitrogen in the soil (Bloom, 2015) and the amount of light the plant receives (Fischer et al., 2015). If the source of nitrogen is from nitrate, the ERa will go up because of nitrogen assimilation. However, as stated in line 597, the main source of nitrogen in Hyytiälä is ammonium. Ammonium does not affect the ERa. If the plants are deprived of light for a longer time, the ERa will also change. As the plants cannot produce enough small carbon compounds from respiration it has to use larger carbon compounds from its storage. However, this would mean that the plant would receive very little light for a longer time period, which did not happen during our measurement campaign. We therefore could say whether the ERa also stays constant during the day.

There are only a few studies that focus on the diurnal variability of ERr and ERa and the studies that do exist mainly focus on the effect of relative abrupt changes in the environment and their effect on the ER signals. It is therefore not yet known whether ERr and ERa stay constant throughout the day and if our assumptions are valid. However, as the existing studies suggest, only major changes in the environment could lead to large changes in ERr and ERa. This means that on diurnal time scale only small changes in ERr and ERa occur, which are too small to measure with our measurement set-up and therefore also fall outside of the scope of this research. Further research using soil, plant, branch or lab chambers is also recommended to obtain further knowledge about the variability of these process level ER signals.

**We have updated the text to include further discussion on this point. The exact changes, together with their line numbers can be found in the list with notes from the pdf, which is included below.**

3.  The data were compromised at times by the failure of some MKS pressure/flow controllers.  The authors apply a correction to the data, but there are a few points with (apparently) anomalous values where we're told that the correction simply wasn't adequate.  Since we aren't told any of the details of the correction, I'd like to see evidence that the other (non-anomalous and corrected) data are valid, and not just because their values are close to what we expect.

    We agree with the reviewer that we did not discuss well enough how the pressure corrections influenced our data and why the uncertainty of this correction increased during the mid-day measurements. Therefore, we added an additional figure to the appendix of the paper and elaborated more on this topic in the text. A more detailed explanation of our correction and the changes we made to the text can be found in the response to reviewer #1, at the first major point, and changes to the text in lines 164, 320, 332.

4.  The authors attribute differences between $ER_{atm}$ and $ER_{forest}$ to "boundary layer dynamics and entrainment" or the unique nature of boreal ecosystems.  I think the first explanation misses the point and the second if very likely wrong.  Whenever you see O2 and CO2

changing with time with a slope more negative than -1.2, this indicates the influence of fossil fuel combustion.

We agree with the reviewer that our phrasing of the explanation for the difference between ERatmos and ERforest and the effects of entrainment should be made clearer in the text. We understand that the ER of fossil fuel combustion has values than 1.2, however we disagree with the point that a measured ERatmos signal higher than 1.2 should automatically indicates a source of fossil fuel combustion. We calculated the footprint for the representative day (7 through 12 July), and it shows that the dominant wind direction is from the North to Northeast, where hardly any sources of fossil fuels are located.

The large value for ERatmos, especially during P2, is a point of concern for all the reviewers. We realize that we did not explain our reasoning well enough to show in a clear way how this large number of around 2 could arise. A more elaborate explanation is given in the answer to reviewer #1 in the section "Explanations for the high ERatmos values". Please see our response to reviewer #1, and our updates to the text in line 481.

We will elaborate here on the specific points that reviewer #3 made in the pdf annotations related to this major point:
Air that is entrained from the residual layer and the free troposphere is influenced by air masses with different background signals. This means that different sources, such as ocean, fossil fuel and biosphere could have contributed to this signal. Mixing different sources of air with different ER signals could create a mixture of air that has a final ER that cannot be contributed to one specific process or could even have an ER value that is higher than 2.

This happens because we cannot just average each ER signal that is mixed into the free troposphere, as an ER signal of fossil fuel has a different meaning than an ER signal of the biosphere. The ER signal of fossil fuel means that the air is depleted of $O_2$ and enriched of $CO_2$. The ER signal of net biosphere exchange means that the air is enriched with $O_2$ and depleted of $CO_2$. Averaging the ER signals would then be wrong, similarly to averaging the day and night ER signals of a forest (see our reply to point 3 of reviewer #1). Different sources contribute to the air in the free troposphere differently for $O_2$ and $CO_2$, it is therefore highly unlikely that the ER signals from different background sources can still be distinguished.

During the morning transition this air is entrained and measured by our devices. When different ratios of $O_2$:$CO_2$ are entrained in our aggregate day, we find this steep slope of around 2 during the entrainment dominant period. The 125 m measurement height also confirms that entrainment is causing these high values. At 125 m, the measurements are outside of the roughness sublayer and therefore the measurements at this height represent the mixed-layer signals that are normally more influenced by the entrainment processes, which results in a slope of 3.40. The ERatmos value is therefore a useful tracer to quantify how the impact of atmospheric driven processes on the measurements. However, to link the ERatmos signal to only the biosphere processes, we recommend using measurements as close as possible to the canopy.

5. There appears to be circularlity in some of the analysis. For example, the EC data are used to set a value of the free parameter K (a transport coefficient) for getting fluxes from O2 gradients. Then the O2-based fluxes are assessed by comparing them to the EC

data. Similarly, NEE is split into GPP and TER using the O2 and CO2 data. Then the O2 and CO2 data are further interpereted by taking GPP and TER as if they were known a priori.

We agree with the reviewer that there is indeed a minor degree of circularity in our analysis for the 1) flux partitioning and 2) the use of K. We will address these two points and clarify the limited impact on our analysis.

1) The circularity in the $O_2$ method. We agree with the reviewer that it could have been written down more carefully how we calculated the flux partitioning using the $O_2$ method as shown in figure 8. We used the GPP and NEE measurements (that were available from the EC measurements at Hyytiälä) to determine ERa for the representative day (7 through 12 July). Then we applied this ERa to a new set of days (13 through 15 June) using the $O_2$ method. For the new set of days, we used the ERa and ERr that were based on the initial representative day and the ERforest was calculated with the measurements of the vertical gradient and resulting $O_2$ and $CO_2$ fluxes for the new set of days. This means that there is a certain degree of circularity in this calculation, as the ERa is based on GPP measurements from one combination of days (7 through 12 July) and then we use this ERa again to determine GPP for a new combination of day (13 through 15 June). However, we expect only minor changes in ERa compared to the values we derived, and the good comparison between the $O_2$ and the EC methods for the partitioning of the fluxes shows the potential of the $O_2$ method. We further elaborate on this point also in our reply to reviewer #2 and updated the text in line 611.

2) The circularity in the K validation. It is indeed correct that we use the EC-based CO2 fluxes to calculate the K and then use K to derive our fluxes from the vertical gradients and compare them again to the EC-based $CO_2$ fluxes. This results in a circularity which is mentioned in line 517. Unfortunately, this approach cannot be tested on one day (first representative day) and then be applied to a different day (the second representative day), as we did above to check the $O_2$ method.

However, for this flux calculations of $O_2$ and $CO_2$, our approach is again not completely circular, because we use the EC flux of $CO_2$, together with the gradient of the ICOS $CO_2$ mole fraction observations to determine K. Then, we use this derived K, together with the gradient of our campaign data to determine the $CO_2$ flux and compare it again to the EC flux. With this approach, the vertical gradient is measured by two different instruments, which makes this comparison not completely circular. This circularity will always exist even if we would test it on another day which makes it difficult to check the method of using $CO_2$ to determine K without being biased. The circulatory of determining K is therefore difficult to solve.

To remove this circularity, we could choose to use another parameter measured with EC, such as potential temperature and the sensible heat flux to determine K. We have used this approach to test several options for deriving K, as we show in table 2 of the manuscript, and these different approaches can also be used to get an idea of the uncertainty of K and the impact on our results. However, we prefer to use $CO_2$ as it has a more directly linked to $O_2$ on how it is transported through the atmosphere, compared to potential temperature, which is related to the sensible heat flux and therefore an active variable in generating turbulence. Other studies (Wu et al., 2015)

also use $CO_2$ as a measure to determine K and we therefore are confident that this approach works. This is discussed further with point 30 below.

We have included table B1 in the appendix of the paper to clarify the data used in each step of our calculations, so the level of circularity in our calculations is more clearly shown.

It's quite possible (particularly for #5) that the authors have done nothing wrong and I have simply failed to understand their work. If that's the case, then my comments should be taken as a plea for clarification and explanation in the text.

All of these concerns, along suggestions/corrections on word choice, punctuation, sentence structure and grammar, and covered in the attached "marked up" PDF. The markings are in three colors: Red - add/delete/move text, to be taken verbatim   Green - questions/directives for the authors  Yellow - highlighting text for which I have typed a "sticky note". Be sure to open the note and read to the end. Scrolling may be required.

We copied the sticky notes comments from the PDF file below, with the line number they were attached to, so we could provide answers with each comment. Finally, we have updated the text using the textual suggestions provided by this reviewer.

Finally, I would like to acknowledge that the writing quality is very high. Even though I have made numerous editorial markings, as a native English speaker (with a modest proficiency in German) I am in awe of the authors' ability to write so well in a second language. Well done!

Thank you for this compliment, we tried to improve the writing further using your suggestions.

1. Line 36: The semi-synonymous use of OR and ER is a problem in this field. OR is a chemical property specific to materials and investigated by elemental analysis or combustion (or similar methods) in the laboratory. ER is a behavioral "symptom" that is specific to an organism, group of organisms, or ecosystem. Then we have alpha_b, which is an effective ER for the planetary biota and has things like disturbance and wildfire built into it. You should be careful of these distinctions throughout the paper and choose your labels accordingly.

   We agree with the reviewer that there is a distinct difference between the OR, ER and alpha_b, and we should have been more careful explaining them. Therefore, we update the text accordingly and have added an extra sentence in line 57, that explains the difference between these two terms.

2. Line 58: Here again there is the problem of OR vs ER. It's not just a temporal difference - there's also a spatial scale. OR from elemental analysis is intrinsically tied to particular

samples. Leaves may be different from twigs or trunk wood, and you only learn about what you put in your analyzer. Your prose should be chosen with great care to reflect these distinctions.

We agree with the reviewer that the difference between OR and ER is not just based on temporal difference, but also on spatial. Therefore, we have added extra details to line 58 to make this more clear.

3. Line 60: ...and here, alpha_b should be reserved for a single global number. The budget equations using alpha_b are really only meaningful on large scales, so we can't talk about local values of alpha_b. + Line 79

We agree with the reviewer that the alpha_b that is used on large scale, e.g. to estimate the ocean sink, should be taken as a single global number that account for all the specific processes. However, when the O2 method is applied on a more local scale, for example using the APO method to determine fossil fuel emission (Pickers et al., 2022) a more local $O_2$:$CO_2$ molar ratio for the biosphere has to be used, that indicates more the processes that influence the measurement location. In this way we could talk about local values for the biosphere exchange. We therefore have removed the reference to alpha_b when talking about a more local scale.

4. Line 66: If you want the ER of the forest, any real measurements of "surface" fluxes won't work (unless "surface" includes the soil surface, leaf surface, trunk surface, petiole surface, etc.). I believe you're imagining an idealized surface, so you should probably say so explicitly.

To be clearer with what we mean by surface, we added some text to line 64.

5. Line 78: Seibt et al's paper is a wonderful one, but they did not present continuous measurements of air above a forest- she analyzed discrete flask samples. Probably better to say "...that measure O2 and CO2 in the atmosphere above an ecosystem with sufficient frequency to derive ER values."

We agree with the reviewer that we should have been clearer about which measurements Seibt et al. (2004) did and therefore we changed the text in line 78.

6. Line 175: I'm probably missing something simple, but it's not clear to me why a different Target Tank should yield a different std dev. Is there some simple explanation you can include in the text?

We added a few words to line 175 to explain that the target tank of 2018 and 2019 differ because they contain different composition of air. This results in a different mean difference to the calibrated value. In principle, the std should be the same if the system did not change between the years, and when averaging over a long enough period. However, for our short campaigns we still found some differences between the two years, and therefore report these separately.

7. Line 191: Presumably you mean a negative relationship in the changes of O2 and CO2 as a function of time (i.e. dO2/dt and dCO2/dt have opposite signs). Please clarify.

We changed the line 191 to make clear that the negative relationship indeed indicates a relationship of changes over time.

8. Line 225: I think what you're trying to say is "the flux for any give entire day is the average of the fluxes for the unstable (daytime) period and the portions of the stable (nighttime) periods that lie in the chosen midnight-to-midnight 24hr window. The data from the transitional periods (excluded from the day and night periods) are not included in the full-day average." Maybe I have misunderstood what you are saying, but if so, that only strengthens my point: Please consider trying to make this clearer, rather than just saying "the flux for the entire day is the average over the entire day".

We added some text to line 225 to make clear that we use all the points to calculate the average fluxes of $O_2$ and $CO_2$ to determine the $ER_{forest}$ of the entire day. The transition periods are now also included.

9. Line 258: I trust that you're not actually doing anything wrong here, but this whole section ER_forest is a bit confusing since I can't clearly tell when you switch from discussing the observational approach to the theoretical one. In particular, in the observational method, you need EC-based CO2 fluxes to calculate K. Thus, "choosing the value of K that gives the best agreement with the observed EC CO2 fluxes" is circular. I don't actually think you're falling into this trap, but I hope you can see how this existing prose might be confusing to the reader. Please try to reorganize the content so that the two approaches are more clearly distinct.

For a more elaborate explanation on the circularity, please look at major point 5, and below at point 29.

We added some text to line 258 to make it more clear that we test the best approach to calculate Kφ based on both the observational and theoretical approach.

10. Line 272: This assumption is essential for your analysis that follows, and it's probably correct enough (given the uncertainties in your O2 measurements). However, I think you should consider and acknowledge the possibility that respiration during the day (which includes photorespiration, particularly in hot, dry conditions) may actually have a slightly different ER_r than respiration at night. I'm no plant physiologist, but I'd like to know that you have at least thought about photorespiration and its potential link to nitrogen in the tissues (e.g. see Bloom, Photosyn. Res. 123(2):117-128) and how it might make ER_r different during daylight hours.

We agree with the reviewer that we should explain our assumption that ERr stays constant throughout the diurnal cycle. Please see our reply above for major point 2 for a detailed explanation on why we assume that ERr stays constant during the day and night and please look at line 272 for the changes made in the text.

11. Line 279: In Fig. 3 you only shade July 7-12 and it's quite hard to see anything about July 13-15. Please shade these dates too (but with a distinctive tone) and include them on the zoomed-in plots. I would like to be able to assess whether the data really do look comparable for the two intervals.

We added a shade to the days 13-15 June to make it more clear which days we choose for the analysis and calculation of the $O_2$ method. The zoomed-in plot of these particular days was added to the appendix and show here below in figure 1.

[Figure]

*Figure 1 The O2 and CO2 measurements at 23 m, and the vertical gradients of O2 and CO2 for the second representative day (13 through 15 June 2019). This is added to figure 3 in the paper.*

12. Line 288: This is a clever approach, but it only works if you have ER_a and ER_r in hand and you are confident that they are the same on the day you determine them and the day you use them to get GPP and TER. That seems like a good assumption in this case, where your two "representative days" are not very far apart (temporally and spatially), but I would like to see some prose addressing this assumption/limitation.

We agree with the reviewer that we should elaborate more on why we justify the assumption that the ERa and ERr of the period 7 through 12 July can be used for the period 13 through 15 June and can therefore be used to calculate GPP and TER from the $O_2$ method. As both periods are relatively close to each other and have similar meteorological conditions, we can assume that the ERa and ERr stay constant based on the few studies that exist on this topic (see above our reply on major point 2 for further explanation). **We have added more elaborated information to the text in line 281.**

13. Line 305: This is valuable information. Could you please mark it (with vertical dashed lines, or something like that) in Figure 4?

We included the sunrise and sunset times with vertical dashed lines in Figure 4. Be aware that Figure 4 is in local wintertime and the time that is given in the text is summertime. This means that there is 1 hour difference.

14. Figure 4 caption: "A remaining artefact" is too vague. My immediate reaction was that you were talking about the single point that jumps up. However, after reading the body of the text, I see that you're referring to a bigger problem. In this caption, you should either omit any reference to the problem or highlight the effected points on the plot and say "see text for details".

We agree with the reviewer that we should refer to the text to highlight that this period was part of a bigger problem. Therefore, we adjusted the text in the caption of figure 4.

15. Line 319: I think you're saying that you expect the O2 concentration to remain high since you have every reason to believe that assimilation is dominating. However, O2 falls in your plot and you're saying this is due to the MKS problems that you couldn't correct for. If

I've interpreted your writing correctly, this explanation is plausible, but it does make me wonder about the robustness of the rest of your data. Why would the MKS problem be irreparable only in the late afternoon? And since this is a composite "day", are you saying that it was an irreparable problem only in the late afternoon on several consecutive days?

The reviewer indeed interpreted the text correctly and we indeed meant that $O_2$ should have stayed high, but due to the high uncertainty of the PMKS correction during the day the pressure correction could not account completely for the MKS problems. This happens during the day because the pressure instability issue has a high correlation with temperature.
A more elaborate explanation about the PMKS correction and why it mostly affected the mid-day data is given in the answer to the major point raised by reviewer #1. Please see for more detail the section "Further details on MKS pressure transducer instability correction". We have modified the text in lines 164, 320, 332 to explain this in more detail.

16. Line 326: I'm sure I'm just missing something obvious, but you say the CO2 gradient goes from negative to positive due to CO2 being transported downwards. Wouldn't that have the opposite effect?

The gradient is calculated by subtracting the observations at 23 m from the observations at 125 m (125 m-23 m). This means that for example when $CO_2$ has a positive gradient during the day, the concentration of $CO_2$ at 125 m is higher compared to concentrations at 23 m, since the 23 m is closer to the sink of $CO_2$ driven by photosynthesis which reduces the concentration. This means that the forest is taking up more $CO_2$ to deplete the air of $CO_2$ at the height of 23 m compared to 125 m. As a result, the gradient becomes positive.

As equation 7 also shows, the gradient and the flux are anticorrelated. Which means that with a positive gradient ($CO_2$ during the day), the flux is negative and points downwards into the forest.

17. Figure 5: To me, this looks rather like two different "regimes" within P2. As you say in the text, these slopes are probably not telling us about the forest, but instead changes in the degree to which entrainment and other physics is driving the concentrations. It makes me wonder about the choice of boundaries for the various time periods. I can see these points most clearly between 4am and 6am in Fig. 4. One possibility is that the stability of the layers close to the surface (i.e. around 25m) starts to break down before the plants wake up and start photosynthesizing vigorously. Is your micrometeorological data consistent with this?

We agree with the reviewer that there indeed seems to be two periods inside P2. The steep increase is very likely caused by entrainment or other processes which are dominant and that are not related to the surface fluxes. Therefore, it is still correct to assign this period to P2, as we define this as the entrainment dominated period. We are currently working on a follow up modelling study where we aim to further investigate this pattern and try to understand how this steep increase in the morning is formed.

18. Figure 7: There is probably a simple explanation, but I am surprised that the error band for this period is so small, given that the spread in the individual points appears larger than the spread in the 0:00-4:00 period. Perhaps this is worth addressing in the text or in the figure caption.

We used the same uncertainty for the whole night period, as ERr is based on all the night measurements. Therefore, is the error band the same size for the period between 22:00-00:00 as the period 00:00-04:00. The reason why the individual points show larger variability between 22:00-0:00 is because the sun then had just set and the gradient of both $O_2$ and $CO_2$ is still relatively small compared to the period 00:00-04:00. The smaller gradient results in a larger uncertainty in determining the surface flux and therefore a more variable ERforest.

The error band therefore is representative for all nighttime measurements rather than for individual points.

19. Line 396: This is just a confusion over choice of words, but I think your "averaged fluxes" are what I would call "a flux-weighted average of the measurements". I think you are saying that you did such a flux-weighted average for each of the periods in the day, as well as the full 24-hour day. This is particularly confusing for me because all of the really big spots in Fig 7 (i.e. the times with large fluxes) have ER_forest values that are more negative than -1.0 and the biggest are more negative than -2.0, yet your full-day value is only -0.83.

We agree with the reviewer that our method could be described as a flux-weighted average. However, this applies only for the daytime and nighttime signal and not the 24 hours signal. We give an example with the figure below where we calculated the mean of all the ER data points of the daytime. The mean of the ER signals (red line) is lower compared to our daytime ER signal, which is based on the average $O_2$ flux divided by the average $CO_2$ flux (blue line) (representing the flux-weighted average). By using the averaged fluxes, we use indeed a flux-weighted average, as our daytime ER signal is higher compared to the mean of the ER signal. This means that the higher surface fluxes, with a higher ER are given more weight.

We understand that it can be confusing if we look at the 24-hour value, which is 0.83. This is because the daytime and night-time ER represent opposite fluxes of opposite sign for day and night for both $O_2$ and $CO_2$ and therefore influence the atmosphere composition differently. We cannot apply a weighted average when combining the daytime and night-time as we would then ignore that the ER represents different and contrasting processes (night respiration versus day photosynthesis and stable stratified conditions versus convective conditions) during the night compared to the day. For a more detailed explanation please see our answer to point 3 of reviewer #1. We have modified the text in lines 398 + 552 + 559.

[Figure]

*Figure 2 The same data points as Figure 7 in the paper, but now comparing two approaches to calculate the average ER for the daytime: Using the average of the ER signal (red line) and using the average O2 flux divided by the average CO2 flux (blue line).*

20. Line 404: Am I correct that this calculation assumes that you have values for NÉE and GPP from the EC measurements taken by SMEAR II?  If so, you should say so explicitly. Please convince me the rest of your data are properly corrected.

    The reviewer is indeed correct that the values for NEE are from the EC measurements of SMEAR II, and the GPP values are derived from these NEE EC measurements. To make it more clear how GPP is determined from NEE, we have added the reference of Kulmala et al. (2019) and more **explanation in line 275.**

21. Line 405: See comment line 272

    Please see our reply to major point 2 above for a detailed explanation and for the changes in the text in lines 405 + 585 (see also our reply to point 32 below).

22. Line 411: I think I missed something.  Have you already shown (derived/proved) this, or have you just stated it?  I don't think you actually show it in the text that immediately follows either.

    See next point for clarification.

23. Line 412:  Actually, my confusion is more profound than just my "show" comment above. Eqns 7 & 8 have six separate terms.  If you have NÉE and ER_forest and ER_a and ER_r then you can definitely solve for the two remaining ones (GPP and TER).  However, as I understand it, the only way to get ER_a and ER_r is from these equations, using NEE and GPP from independent EC data (from SMEAR II, in this case).  Thus, this looks like a circular (i.e. self-referencing) approach.  The only way around this (which is probably what you're doing) is to calculate ER_a and ER_r for one set of data (i.e. one particular representative day) and then apply it to other data (a different day).  If this is in fact what you're doing, you should be very clear about it.  Otherwise, the calculation appears completely circular.

Please see our reply to major point 5 above and our reply to the second major point of reviewer #2 for a clarification on the reasons why we think there is only a minor circularity in our calculations, because we use different days. We have updated the text in line 611 to explain this further.

24. Line 418: Since you say "increasing/decreasing RE_forest", you should put signs on the associated change in your inferred value of GPP (i.e. if RE goes up, does GPP increase or decrease?). You could do this as simply as including +/- or -/+ as appropriate.

    We agree that this was not clear in figure 8. As GPP and RE are coupled with the equation:
    NEE = -GPP + TER
    This means that when TER increases, GPP has to become more negative to keep the NEE constant in time. To make it more clear we changed the signs, as GPP is normally presented in atmospheric studies as a negative flux (meaning pointing from the atmosphere into the forest and removing $CO_2$ from the atmosphere), we changed Figure 8 and line 418 accordingly.

25. Line 421: How did you settle on this value? I would think that you want to be able to do roughly as well as the EC method. Or perhaps, you'd like to be able to see differences between GPP and TER comparable to those implied by the EC data. Either way, you should state how you settled on 0.05 as your threshold.

    We agree with the reviewer that we should be more clear how we determined the 0.05. We based indeed on the comparison with the EC method. With a suggested precision of 0.05 for $ER_{forest}$, the change in GPP and TER fluxes for the $O_2$ method stays in the range of the GPP and TER fluxes of the EC method. However, to create a more detailed discussion on the comparison between the EC method and the $O_2$ method, a higher precision is probably needed, together with an independent measurement of ERa. A detailed comparison between the EC method and the $O_2$ method was not the main goal of this study (see second comment reviewer #2), and we include this analysis to show the potential of the $O_2$ method, and therefore our suggested precision is 0.05. We added a sentence in line 421 to make it more clear why we choose 0.05, and added further discussion in line 611.

26. Line 479: It is tempting to say simply "atmospheric dynamics and entrainment can give a slope closer to -2" but I think that's a little misleading. When non-canopy air is entrained or advected, that air brings both O2 and CO2 with it. This will change the composition of the air you measure, "moving" down-right or up-left in Fig. 5, depending on the composition of the previous parcel you measured. Most importantly, the "movement" will define a slope that reflects the processes that last changed the new parcel. If you are advecting/entraining air that has only seen biological influences, the slope will always be close to -1 no matter how much O2 and CO2 are removed/introduced. The only way the slope can be close to -2 is if the air that was entrained was heavily influenced by natural gas combustion. In short, your observed values more negative than -1.3 are almost certainly indicating that you are measuring non-local air with a strong fossil-fuel signature. Furthermore, the longer the time over which you include data, the greater the footprint and the more likely you are to have non-local influences. This is consistent with your 24-hour values being much more negative than the shorter periods. When you look at it this way, you see that you have to be very close to the top of the canopy (or within it), and measuring over a short time in order to get ER_atm to be at all comparable to ER_forest. I encourage to you to rewrite this section of your discussion with these ideas in mind.

Please see our answer to major point 4 above and our answer to reviewer #1 in the section "Explanations for the high $ER_{atmos}$ values" for a detailed explanation about this point. We have changed the text in line 481.

27. Line 481 – 487: Unless there is some reason to believe ER_forest is very, very different from -1.05 (i.e. much more negative), a much more likely explanation is non-local influences, as described in my previous note (above). As you yourselves discuss below, the OR of the organic matter in a boreal forest has values not so different from -1.0. Simple calculations show that even when there's disequilibrium between the kind of materials being synthesized and those being respired (e.g. low-nitrogen trunk wood vs. shoots and leaves) the ER will differ from the underlying OR by no more than 10-15%. For this reason, invoking this explanation as a real possibility is inappropriate.

We agree with the reviewer that non-local effects are the reason for the high $ER_{atmos}$ values. As described in our reply to major point 2 above and our answer to reviewer #1, we think that it is entrainment from air of the free troposphere that is causing this high $ER_{atmos}$ signal. The air of the free troposphere is influenced by non-local sources. As already discussed, we therefore adjusted the text in line 481.

28. Line 500: Maybe I've just forgotten where you mentioned this earlier, but I'm not sure exactly what you mean by "flux-gradient" method. Please define/remind.

We added to line 500 the reference to equation 7, which indicates what we mean with the flux-gradient method.

29. Line 517: Yes, this circularity is a problem and deserves more than an acknowledgement. I have already made comments about it on Lines 404 and 414. Your basic idea for using all the available information is very clever, but I believe the only way to make it robust is to use one dataset (the first representative day) to "tune" your method, and a different dataset (the second representative day) for drawing conclusions and deriving results. Otherwise, the results are untrustworthy and the error analysis essentially meaningless (or at least far too complicated for me to imagine/understand/trust). Perhaps you've already done this and I just missed it. If so, please make it much more explicit.

Please look at our reply to major point 5 above for an elaborate discussion on this circularity issue and why we think it is not a major issue. We have added information in line 517.

30. Line 536: Maybe I just missed it, but I don't see any quantitative statement about the uncertainty in K and there's no sign of it in Table 2. Without that number, I'm not convinced that the error bars on the nighttime fluxes are really as small as those pictured. And of course, quantifying uncertainty in K ties back in to my comments about circular reasoning above.

We did not include the uncertainty of K in our calculations, because the final uncertainty in our values is from the measurement uncertainty in $O_2$. As equation 7 shows, to determine K we used the EC flux measurements of $CO_2$ and the gradient of $CO_2$ from the heights 16 m, 67 m and 125 m from the ICOS instrument. This means that all the final uncertainties of our results where ICOS data is included, are probably somewhat higher than currently given. However, as stated in line 574, the largest factor of uncertainty in our observations

is from the $O_2$ measurements themselves, which means that probably most of the uncertainty range is captured in the values we present.

To better quantify how the variability in K would effect the final results of the ER signals we test what happens when we use K with theta, instead of K with $CO_2$ , to determine our final CO2 and O2 surface fluxes (see Figure 2 below).

[Figure]

*Figure 3 The same as Figure 6b in the paper, but now with the approach added where the O2 flux is determine with the K based on potential temperature.*

If we use K with theta to calculate the surface fluxes of $O_2$ (Figure 2 above) and $CO_2$ (as for Figure 6a in paper), we find that the difference with the approach to use $CO_2$ for K is minimal (red and green data points in the figure 2 above). The largest differences arise in the transition periods. This is expected, as the gradient changes fast during this period and differences between the gradient change in theta compared to $CO_2$ are then more pronounced.

If we would then use the fluxes based on the K with theta, we get the following ER signals:

*Table 1 The ER signals, same as Table 3 in paper, but now with the approach where K with theta is used to determine the surface fluxes. For reference the ER signals of the approach where K with CO2 is used is also added.*

| ER signal | Value (based on theta) | Value (based on CO2, from table 3) |
|---|---|---|
| ERforest_day (09:00-17:00) | 0.94 | 0.92 |
| ERforest_night/ERr (21:00-04:00) | 1.04 | 1.03 |
| ERforest_all (all data point) | 0.92 | 0.84 |
| ERa_day (09:00-17:00) | 0.97 | 0.96 |
| ERa_all (all data points) | 0.99 | 0.96 |

We find by comparing the table above with the ER signals based on $CO_2$ from table 3 (also included in the table above), that most of the ER signals are quite similar. The biggest difference can be found between the ER signals where the transition periods are included (ERforest_all and ERa_all), which was expected as the transition periods were most difficult to determine.

By using different methods to determine K, we showed that even with a variable K the final results are still very similar. This shows that we can be quite certain about the derived K and the uncertainty of this component is low. As already stated above, the largest uncertainty is caused by our $O_2$ measurements and therefore it is reasonable to omit the uncertainty in the K from our uncertainty calculation.

To make it more clear that we could not include the uncertainties of the ICOS data we added some text in line 543.

31. Line 565-577: This entire section ties into my comments on Lines 478 and 485. Please revise accordingly.

As described in our reply to major point 4 above and our answer to reviewer #1, we think that entrainment is the largest contributor to the high ERatmos signals.

We added to line 466 in more detail why entrainment is the cause of this large difference between $ER_{forest}$ and $ER_{atmos}$ and that our measurement height is probably a first explanation why we find such different results compared to previous studies.

32. Line 583-592: I already commented extensively on this at Lines 272 and 407. To add something specific here: I am not a plant physiologist or soil scientist, but I have the distinct impression that respiration is distributed through the ecosystem, so such a close focus on soils may be myopic. I am quite willing to change my opinion about the primacy of soil composition, but I would like to see some evidence that root/trunk/foliar respiration is a second (or third) order concern, particularly in the daytime.

We agree with the reviewer that we should have been more clear in our explanation on why we think that the ERr is mainly formed by the soil. To elaborate further on this, we added more details in the text in line 585 and refer to the study of Hilman et al (2022).

33. Line 594: Perhaps I'm confused, but since ER_r is assumed to be the same day and night, then ER_a is zero at night. This means the comparison is really day-only vs. day-only + transitions. Perhaps you should present the two methods this way (rather than "day" and "all day").

We agree with the reviewer that our ERa for 24 hours is also the ERa with the day-only + transitions, as the ERa is zero during the night. We decided to present the ERa as all day/24 hours instead of day-only + transitions to be more certain of our final ERa value. The transition periods are the most uncertain as here the gradient becomes very close to zero and even switches sign. To make sure that the average NEE and TER fluxes are more robust we therefore included also the nighttime measurements. For GPP it does not matter as the GPP flux during the night is zero. For the ERa calculations, we need the NEE, TER and GPP fluxes. When these fluxes can be calculated with a lower uncertainty, the uncertainty of ERa also goes down. In principle the ERa should be the same between day + transitions and 24 hours and therefore it is better to choose the method that gives the lowest uncertainty.

Next to that, when deriving this value of the ERa for the entire day it is hopefully clear that this ERa is the ERa for this ecosystem throughout the diurnal cycle and can therefore be used for the $O_2$ method, presented in Figure 8. It links better with the ERforest of the entire day and hopefully makes it easier to understand how all the ER signals are linked.

34. Line 600: Unless there's a typo, the two methods give results that are identical (not just close), and you couldn't tell them apart even if they had tiny uncertainties.

We agree with the reviewer that both the ERa calculations have the same result. After adding some references, we now expanded our explanation that we can assume that ERa stays constant over the day. As other studies shows that the ERa only changes with major environmental changes, which did not happen during our period of days used of the representative day.

35. Line 630: See comments above (lines 478 and 485)

We think the word 'entrainment' is right here. For a more detailed explanation please see our reply to major point 4 above and our answer to reviewer #1.

**Citation**: https://doi.org/10.5194/acp-2022-504-RC3

References:

Angert, A., Yakir, D., Rodeghiero, M., Preisler, Y., Davidson, E. A., & Weiner, T. (2015). Using O 2 to study the relationships between soil CO 2 efflux and soil respiration. *Biogeosciences*, *12*(7), 2089–2099. https://doi.org/10.5194/bg-12-2089-2015

Bloom, A. J. (2015). Photorespiration and nitrate assimilation: A major intersection between plant carbon and nitrogen. *Photosynthesis Research*, *123*(2), 117–128. https://doi.org/10.1007/s11120-014-0056-y

Fischer, S., Hanf, S., Frosch, T., Gleixner, G., Popp, J., Trumbore, S., & Hartmann, H. (2015). Pinus sylvestris switches respiration substrates under shading but not during drought. *New Phytologist*, *207*(3), 542–550. https://doi.org/10.1111/nph.13452

Hilman, B. (2022). *The Apparent Respiratory Quotient of Soils and Tree Stems and the Processes That Control It Journal of Geophysical Research : Biogeosciences*. https://doi.org/10.1029/2021JG006676

Kulmala, L., Pumpanen, J., Kolari, P., Dengel, S., Berninger, F., Köster, K., Matkala, L., Vanhatalo, A., Vesala, T., & Bäck, J. (2019). Inter- and intra-annual dynamics of photosynthesis differ between forest floor vegetation and tree canopy in a subarctic Scots pine stand. *Agricultural and Forest Meteorology*, *271*(February), 1–11. https://doi.org/10.1016/j.agrformet.2019.02.029

Pickers, P. A., Manning, A. C., le Quéré, C., Forster, G. L., Luijkx, I. T., Gerbig, C., Fleming, L. S., & Sturges, W. T. (2022). Novel quantification of regional fossil fuel CO 2 reductions during COVID-19 lockdowns using atmospheric oxygen measurements. In *Sci. Adv* (Vol. 8). https://icos-cp.

Pries, C. H., Angert, A., Castanha, C., Hilman, B., & Torn, M. S. (2020). *Using respiration quotients to track changing sources of soil respiration seasonally and with experimental warming*. 3045–3055.

Seibt, U., Brand, W. A., Heimann, M., Lloyd, J., Severinghaus, J. P., & Wingate, L. (2004). Observations of O2: CO2 exchange ratios during ecosystem gas exchange. *Global Biogeochemical Cycles*, *18*(4), 1–18. https://doi.org/10.1029/2004GB002242

Wu, Z. Y., Zhang, L., Wang, X. M., & Munger, J. W. (2015). A modified micrometeorological gradient method for estimating O3 dry depositions over a forest canopy. *Atmospheric Chemistry and Physics*, *15*(13), 7487–7496. https://doi.org/10.5194/acp-15-7487-2015